# Performance of a prestressed efficiently prefabricated beam-column connection

Jie Cai[1☯¤], Yunlin Jiang[1☯]*, Shuyang Li[2]

**1** School of Civil Engineering, Architecture and Environment, Hubei University of Technology, Wuhan, Hubei, China, **2** Beijing Glory PKPM Technology Co., Ltd., Beijing, China

☯ These authors contributed equally to this work.
¤ Current address: Postgraduate, Dept. of Civil Engineering, Hubei University of Technology, Wuhan, Hubei, China
* jyl115@hbut.edu.cn

## Abstract

Analyzing the seismic performance and flexural capacity of beam-column joints is crucial in structural design phase. The purpose of this paper is to investigate the seismic performance and flexural capacity of precast prestressed efficiently fabricated frame (PPEFF) joints. Reverse cyclic load tests and flexural capacity analysis are conducted. The damage modes, hysteresis curves, skeleton curves, stiffness degradation, ductility, and energy dissipation capacity of five PPEFF joint specimens with different reinforcement rates of the energy-dissipating bars and shear reinforcement are obtained. The results show that the damage pattern of the specimen is ideal, i.e., the plastic hinge region at the end of the beam is severely damaged, whereas the remainder of the beam is slightly damaged. Increasing the reinforcement rate of the energy-consuming steel bars enhances the load capacity, energy dissipation capacity, and initial stiffness of the joint but reduces the ductility performance. The maximum change in ductility was 5.31 for the reinforcement rate of energy-consuming steel bars ranging from 0.38% to 0.59%. In addition, the flexural capacity of the PPEFF joint is evaluated, considering the influence of the shear steel on the yielding and ultimate states. An equation of the flexural capacity is derived. A good agreement is observed between the experimental and calculation results, verifying the correctness of the proposed flexural capacity equation.

## 1. Introduction

Research on precast prestressed efficiently fabricated frame (PPEFF) systems has made substantial progress in recent years. Scholars have conducted experimental studies and applied the results to engineering applications, such as PPEFF frame structure tests, PPEFF beam-column joint studies, and the Wuhan Concentric Flower Garden Kindergarten Project. It was found that the PPEFF system possesses excellent energy dissipation capacity, ultimate load capacity, deformation capacity, and self-resetting performance [1]. Prestressing methods have been investigated and used in different fields and industries, such as steel bridges, timber joints, profiled structures, and braced frames [2–9].

**Data Availability Statement:** All relevant data are within the paper and its Supporting Information files.

**Funding:** The paper funded by the "Industrialized application of BIM-based integrated design

technology for assembled buildings (2020ZLSH08) SL".

**Competing interests:** No competing interests exist.

Scholars from various countries have conducted in-depth studies to optimize the seismic performance of prestressed beam-column joints. In the early years, the research direction mainly focused on the connection type at the beam-column joints, such as longitudinal steel bars attached symmetrically to the beams, to evaluate the energy dissipation capacity and deformation resistance of the joints [10]. In recent years, research on prestressed joints has intensified, and the use of dampers, bolts, and high-performance materials has become more common [11–13]. Some scholars have focused on different parameters of the joints, i.e., the initial prestress magnitude, collapse resistance, prestressing reinforcement with/without bonds, and concrete strength variation [14–17]. Thus, the selection of suitable materials and connections is essential to improve the seismic resistance of beam-column joints. In addition, reinforced concrete (RC) beams have been analyzed extensively using experiments and simulations [18–23].

In the early years, a joint research program of the United States and Japan on prefabricated structures analyzed a hybrid frame system with symmetrical energy-consuming steel bars. The systems had good seismic performance. The research results were incorporated into the design codes of the United States and New Zealand [24, 25]. Due to the complexity of construction and high-quality requirements, the hybrid frame system has not been widely used. When energy-consuming steel bars are installed only in the upper part of the beam, the energy-consuming steel bars near the beam-column interface of the joint are prone to brittle fracture under small displacement [26]. This problem can be solved by applying no bonding treatment of the energy-consuming reinforcement at the beam-column interface. Chinese scholars have investigated beam-column joints with asymmetrical reinforcement. It was found that beam-column joints with prestressed reinforcement installed in the middle of the joint and energy-dissipating steel bars in the upper part of the beam have good energy dissipation capacity and suffer little damage [27–29]. However, due to the complex structure and the lack of theoretical support for shear design, this configuration has not been widely accepted by the engineering community. The China Construction Group has collaborated with many scholars to optimize the design and developed the PPEFF to promote beam-column joints with asymmetrical reinforcement. Prefabricated laminated beams were used to improve the assembly efficiency of the frame and facilitate the installation of the upper beam reinforcement. The energy-consuming reinforcement was partially unbonded to avoid brittle fracture. In addition, shear reinforcement was installed in the upper part of the beam to meet the requirements of the design code for shear resistance. The research results show that PPEFF joints are more efficient in construction, less environmentally polluting than traditional reinforced concrete beam-column joints, and perform better in terms of initial stiffness, bearing capacity, deformation capacity, and damage conditions. In addition, the prestress of the joints was released to verify their shear resistance; the results indicated an excellent performance [1].

At present, researchers have mainly researched the seismic performance of PPEFF joints, but the analysis to quantify the flexural capacity has not been carried out, and there are no corresponding industry guidelines and codes. Therefore, it is necessary to research the flexural capacity of PPEFF joints in order to provide reference for the joints design calculation. In this paper, reverse cyclic loading tests were conducted on PPEFF joints with different energy dissipation and shear reinforcement ratios to evaluate the seismic performance and flexural capacity of PPEFF joints. The damage modes, hysteresis curves, skeleton curves, stiffness degradation, ductility, and energy dissipation capacity of the specimens were compared and analyzed. We established a mechanical model of the beam-column interface and derived equations using the plastic hinge theory to calculate the flexural capacity of the PPEFF joints in the yielding and ultimate phases.

## 2. Methodology

This paper consists of two parts: in the first part, different reinforcement rates for energy-consuming and shear steel bars were used in the reverse cyclic load tests to evaluate the seismic performance of the PPEFF joints.

In the second part, we derive the equation of the flexural capacity of the PPEFF joints. The mathematical relationship between the turning angle and the force is analyzed by establishing a mechanical model of the beam-column interface. The force balance equation and the bending moment balance equation are derived. The calculation results of the equations are evaluated by comparing the calculated and experimental results. Fig 1 shows the flowchart of this paper.

## 3. Experiment

### 3.1. Specimen description and working principle

Five full-size PPEFF joint specimens were designed and fabricated [30, 31]. The dimensions of the beam and column were 300×600×1800 mm and 500×500×2600 mm, respectively. The reinforcement in the beams and columns, except for the prestressing reinforcement, was HRB400 grade reinforcement (denoted by F and D, respectively) and a 25-mm thick protective layer of concrete. The diameters of the energy-dissipating steel bars in the beams of the five specimens were 16 mm, 20 mm, and 25 mm, and those of the shear steel bars were 0, 16 mm, and 20 mm. In addition, the longitudinal reinforcement at the bottom of the beam were 22 mm and 14 mm in diameter to ensure the integrity of the reinforcement skeleton. The details of the specimens are shown in Fig 2. Stirrup rings with a spacing of 100 mm were placed in the beams and columns. Since the fixed end of the beam is susceptible to damage under seismic action, the spacing was reduced to 50 mm in the 500 mm length [31]. The five specimens were referred to as SJ-1, SJ-2, SJ-3, SJ-4, and SJ-5. Table 1 lists the details of the five specimens.

The beam was placed flat on the ground to facilitate the fabrication of the large specimen. Ordinary and prestressed reinforcements were used. The prestressed reinforcement was installed in the middle of the specimen, and the energy dissipating bars and the shear reinforcement was located in the upper part of the beam. High-strength mortar was used to attach the beam to the column. The energy dissipating and shear steel bars in the upper part of the beam were then connected using reinforcement links in the column. A 200-mm long polymeric vinyl compounds (PVC) plastic pipe and a section of heat-shrinkable pipe were installed to prevent brittle fracture of the energy-consuming reinforcement near the fixed end of the beam. A hole was located in the middle of the specimen to facilitate the installation of the prestressed reinforcement. A 200-mm thick concrete layer was poured on the upper part of the beam. Figs 3–6 shows the main fabrication process of the specimens.

When the specimen is subjected to earthquake, the fixed end of the beam rotates around the column, and the unbonded prestressed reinforcement that maintains the elastic tension state improves the self-resetting capacity, flexural capacity and shear capacity of the specimen. When the free end of the beam is subjected to downward load, the beam rotates clockwise, the energy dissipating reinforcement in the upper part of the beam is under tension and the longitudinal reinforcement in the lower part of the beam is under compression, at which time the beam is in the same state of force as the normal concrete beam-column joint. When the free end of the beam is loaded upward, the beam rotates counterclockwise, the concrete in the lower part of the beam is separated from the column, the longitudinal reinforcement in the bottom part of the beam is not stressed, and the energy dissipating reinforcement in the upper part of the beam is compressed. This beam-column opening mechanism can properly

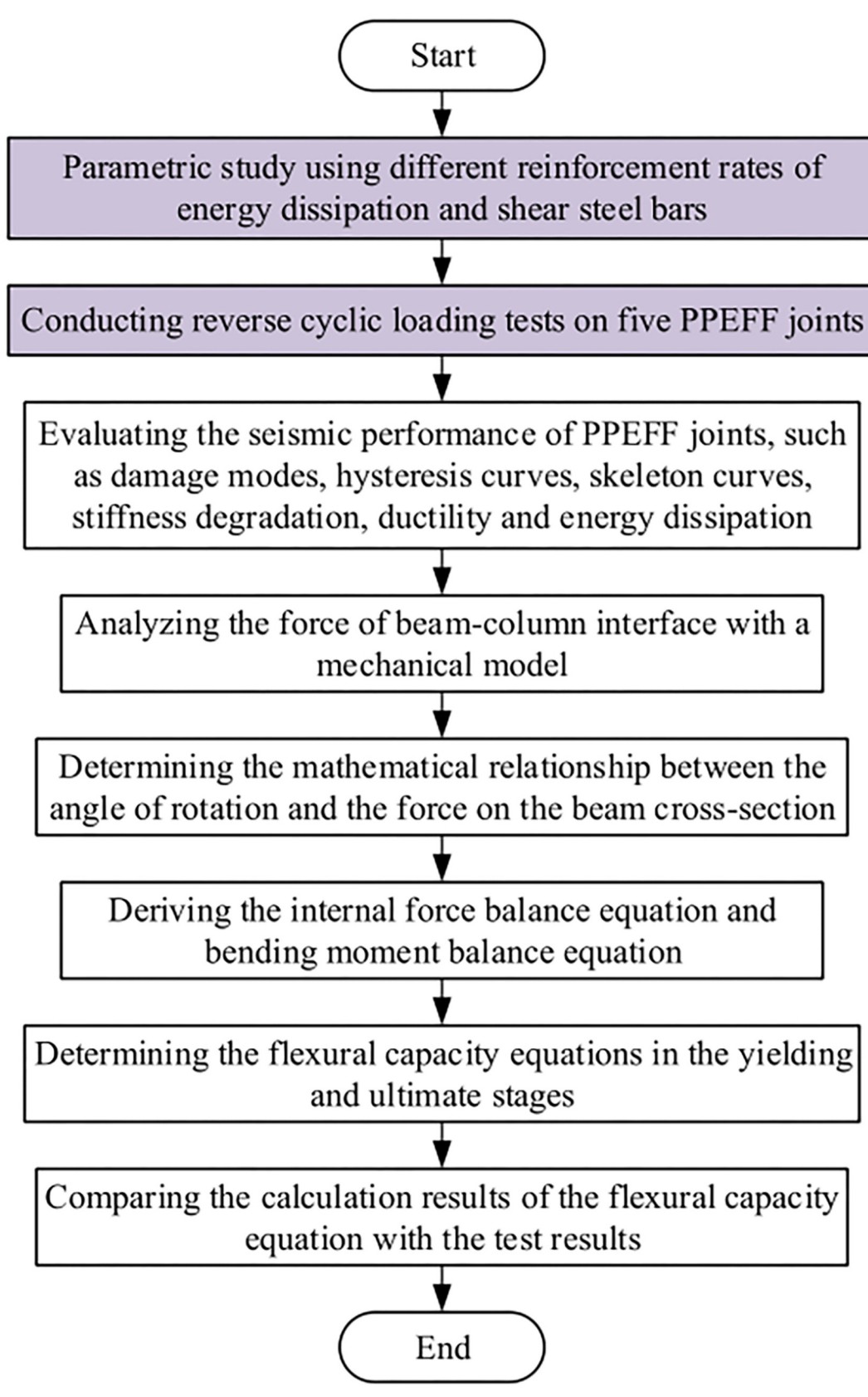

**Fig 1. Flowchart of this paper.**

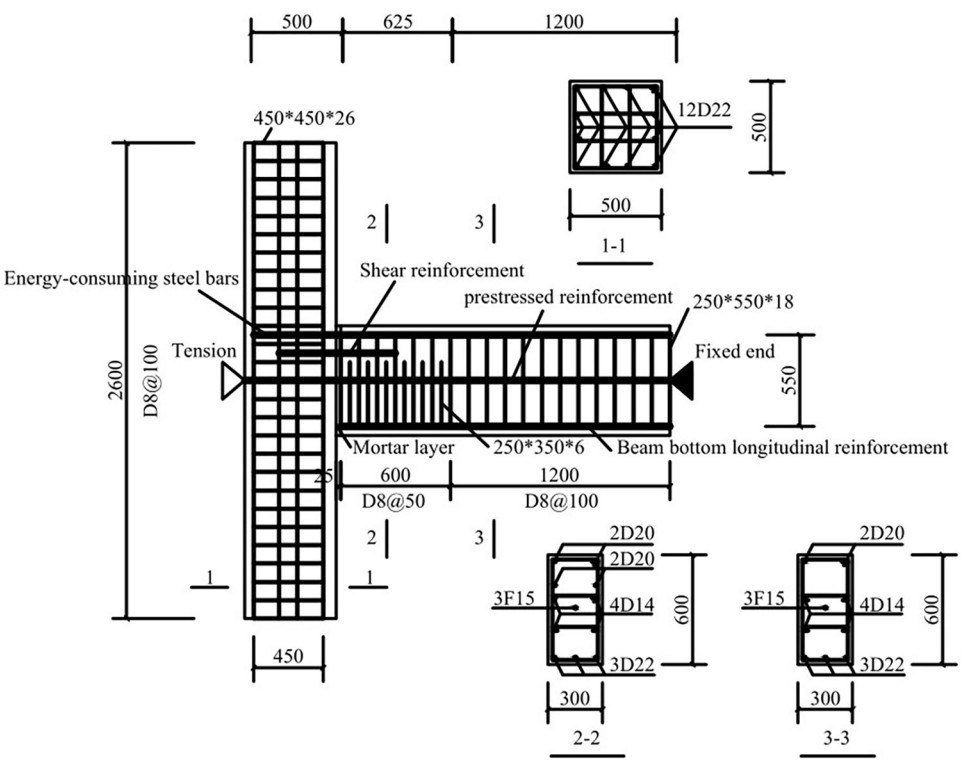

**Fig 2. Specimen reinforcement details (SJ-1 as an example).**

accelerate the emergence of plastic hinge at the beam end, so that the beam end can fully rotate and consume energy to protect the column end, which is in line with the design concept of strong column and weak beam [31, 32].

## 3.2. Material properties

Three concrete cubes with a side length of 100 mm from the same batch were tested to measure the compressive strength of the C40 concrete. The compressive strength and modulus of elasticity of the concrete were 40 MPa and 32,500 MPa, respectively. Similarly, three cubes (side length of 100 mm) of high-strength mortar were tested; their compressive strength was

**Table 1. Information on the five specimens.**

| | Specimen | SJ-1 | SJ-2 | SJ-3 | SJ-4 | SJ-5 |
|---|---|---|---|---|---|---|
| **Beam** | Energy-consuming steel bars | 2D20 | 2D16 | 2D25 | 2D20 | 2D20 |
| | (Area ratio (%)) | (0.38) | (0.24) | (0.59) | (0.38) | (0.38) |
| | Shear steel bars | 2D20 | 2D20 | 2D20 | 0 | 2D16 |
| | (Area ratio (%)) | (0.38) | (0.38) | (0.38) | 0 | (0.24) |
| | Beam bottom steel bars | 3D22 | 3D22 | 3D22 | 3D22 | 3D22 |
| | (Area ratio (%)) | (0.63) | (0.63) | (0.63) | (0.63) | (0.63) |
| | Prestressed steel bars | 3F15 | 3F15 | 3F15 | 3F15 | 3F15 |
| | Stirrups | D8@100 | D8@100 | D8@100 | D8@100 | D8@100 |
| **Column** | Stirrups | D8@100 | D8@100 | D8@100 | D8@100 | D8@100 |
| | Axial compression ratios | 0.15 | 0.15 | 0.15 | 0.15 | 0.15 |

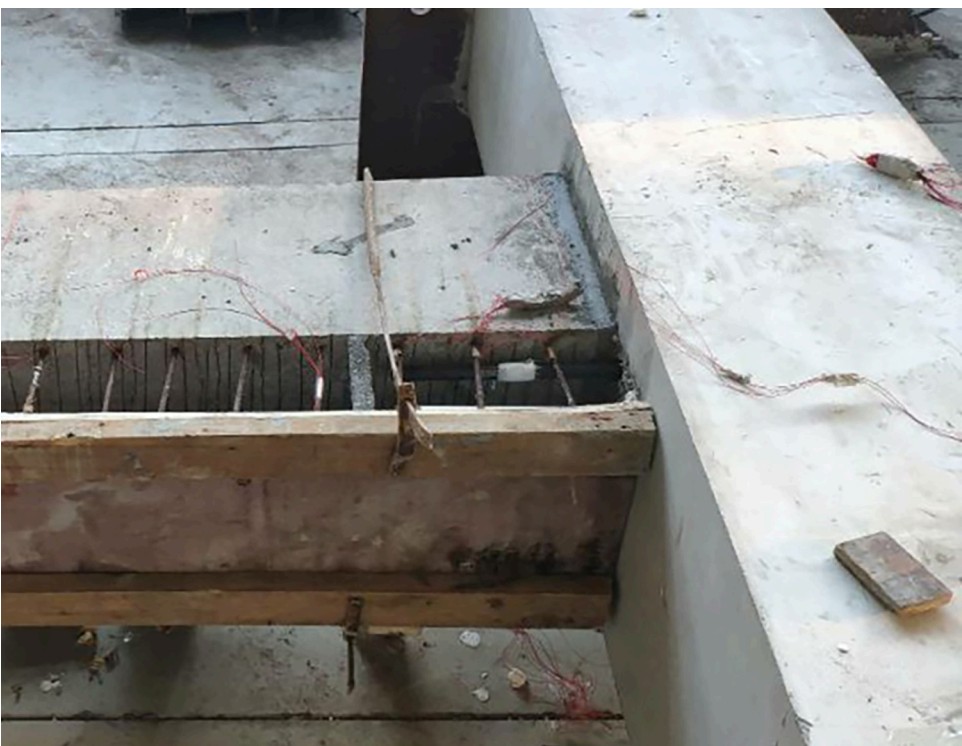

**Fig 3. Pouring high-strength mortar.**

75 MPa. In addition, three pieces of rebar with different diameters were tested. The yield strength and modulus of elasticity of the prestressing bars were 1,986 MPa and 19,800 MPa, respectively [34]. The results of the remaining ordinary reinforcement types are shown in Table 2.

### 3.3. Loading method

As shown in Figs 7 and 8, a 200-t manual hydraulic jack was used to apply an axial force to the column top. The axial pressure of the jack was 717 kN. The bottom of the column was kept stable by welded steel plates, which were bolted to the ground. The fixed end of the beam was attached to the column and remained fixed, while the free end of the beam could move in the vertical direction. Two jacks were used for reverse cyclic loading at the bottom and top of the free end of the beam; the loading rate of the jacks was consistent. The upper and lower jacks were equipped with pressure sensors to obtain real-time measurements. A displacement sensor was placed 60 mm below the free end of the beam to monitor its displacement.

Reverse cyclic load tests are currently one of the most widely used test methods for structural seismic testing. Static loading is used to simulate the force deformation of a structure subjected to seismic effects as a way to evaluate the seismic performance of the structure. The test has a low loading rate, so its effect on the stress-strain of the structure can be ignored, and it is suitable for seismic analysis under different seismic effects. Therefore, reverse cyclic load tests were used to study the seismic performance of PPEFF nodes, which were loaded according to a mixed control of load and displacement [34]. The axial pressure on the top of the column was constant. The loading procedure is shown in Fig 9. It was divided into two phases. Phase 1: Before the specimen yielded, the design yield load $P_y$ was used as the reference, and one cycle with three load values ($0.5P_y$, $0.7P_y$ and $P_y$) was used. A rate of 5 kg/s was maintained.

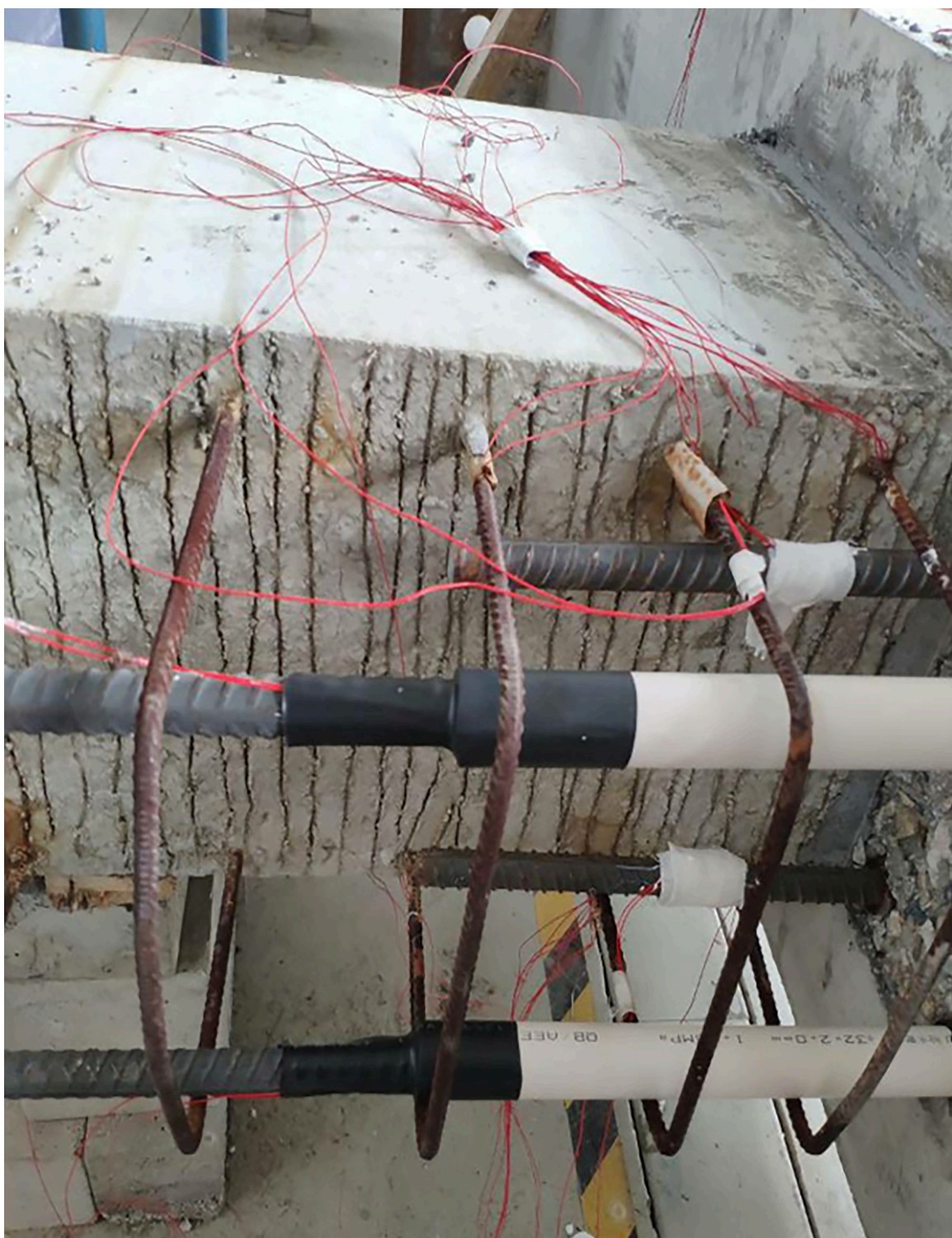

**Fig 4. Installing reinforcement.**

Phase 2: After the specimen yielded, the yield displacement $\Delta_y$ was used. Two loading cycles were completed with an integer multiple of $\Delta_y$ and a rate of 0.067 mm/s. The test was stopped when the specimen was loaded to 85% of the maximum load or was destroyed [24, 33–36].

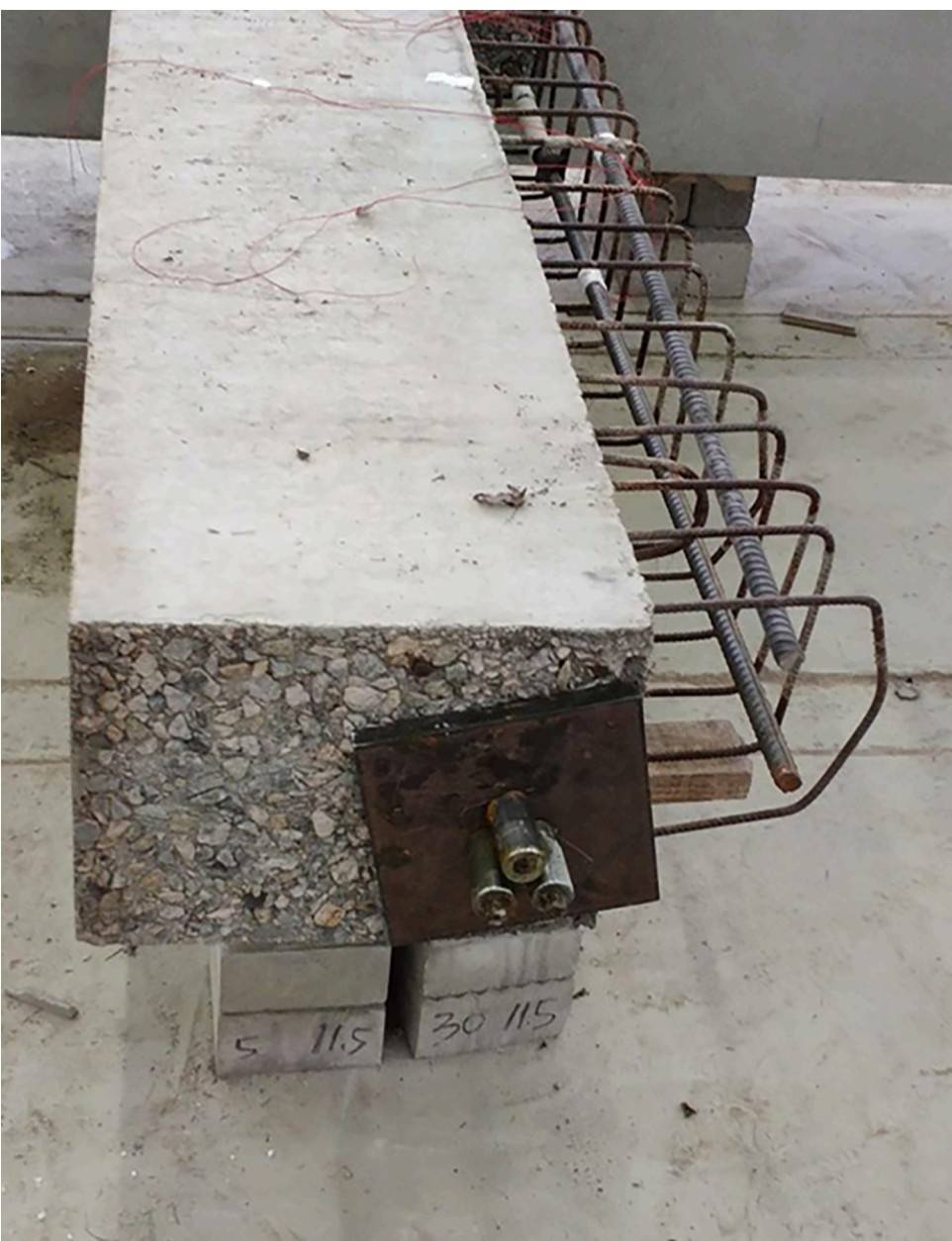

**Fig 5. Installing prestressed reinforcement.**

## 4. Test results and discussion

### 4.1. Test phenomenon

The final damage patterns of the five specimens are shown in Figs 10–15. The failure mode of SJ-1 is shown in Fig 10. The first crack with a width of about 0.2 mm appears at the top of the fixed end of the beam in the phase $P_y$ loading. In the cyclic loading phase of $2\Delta_y$, a major oblique crack with a width of about 3 mm appears at the fixed end of the beam. In phase $4\Delta_y$ cyclic loading, the concrete at the fixed end of the beam is crushed, exposing the energy dissipating bars and shear reinforcement. Fig 11 shows a diagonal crack with a maximum width of 2 mm

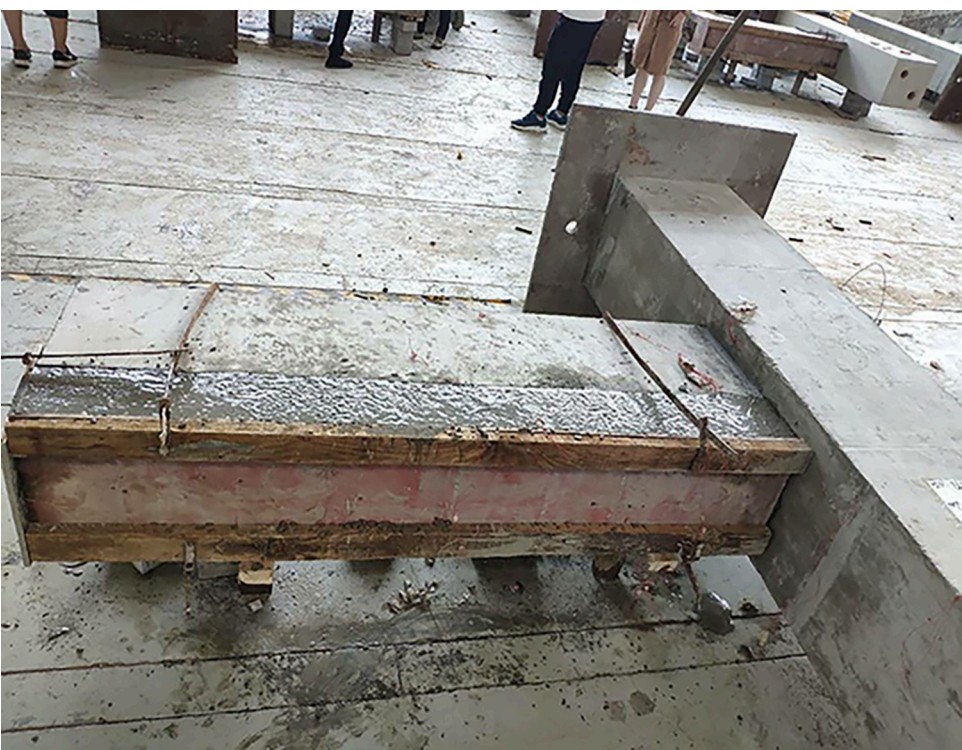

**Fig 6. Pouring concrete.**

in SJ-2 in phase $2\Delta_y$ cyclic loading. Until the end of the test, the damage of SJ-2 was similar to that of SJ-1. Fig 12 indicates that SJ-3 bulges outward, and the concrete is crushed and falls off during phase $4\Delta_y$.

The failure mode of SJ-4 is shown in Fig 13. At the end of phase $0.7P_y$ cyclic loading, a fine crack with a width of 0.05 mm first appeared at the fixed end of the beam. In phase $1\Delta_y$, a thin diagonal crack with a width of 0.1 mm occurred in the concrete at the fixed end of the beam. In phase $3\Delta_y$, the concrete at the upper part of the fixed end of the beam was crushed, and the width of the diagonal crack reached 10 mm. The failure mode of SJ-5 is shown in Fig 14. In phase $4\Delta_y$, the concrete at the fixed end of the beam was severely damaged and dislodged. The energy-consuming reinforcement protruded outward, and the shear reinforcement and hoop bars were clearly visible.

The results showed that the damage of the five specimens was concentrated in the plastic hinge zone of the beams. The damage to the columns consisted of a small amount of crushed surface concrete near the beam-column interface and some minor cracks in the concrete in the core area. The damage in the upper part of the fixed end of the beam was more severe than

**Table 2. Mechanical properties of the ordinary reinforcement.**

| Diameter (mm) | 8 | 14 | 16 | 20 | 22 | 25 |
|---|---|---|---|---|---|---|
| $f_y$ [a] (kN) | 371.25 | 428.86 | 489.83 | 549.27 | 586.48 | 686.59 |
| $E$ [b] ($10^5$kN) | 1.98 | 2.01 | 2.05 | 1.95 | 2.01 | 2 |

[a]the yield load.

[b]the elastic modulus.

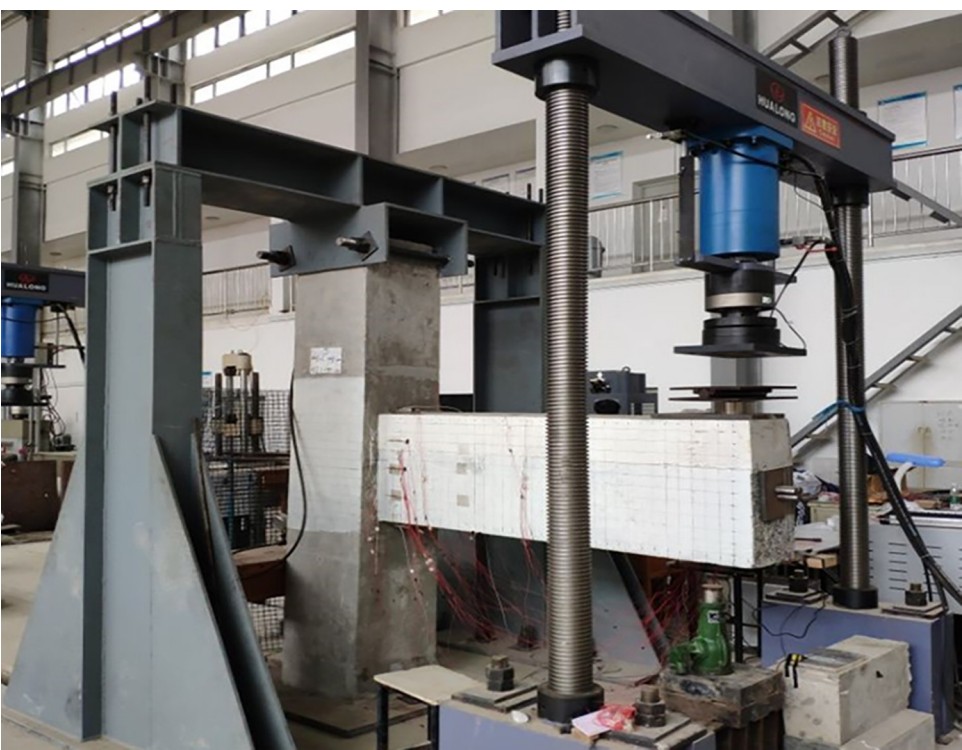

**Fig 7. Test loading (front).**

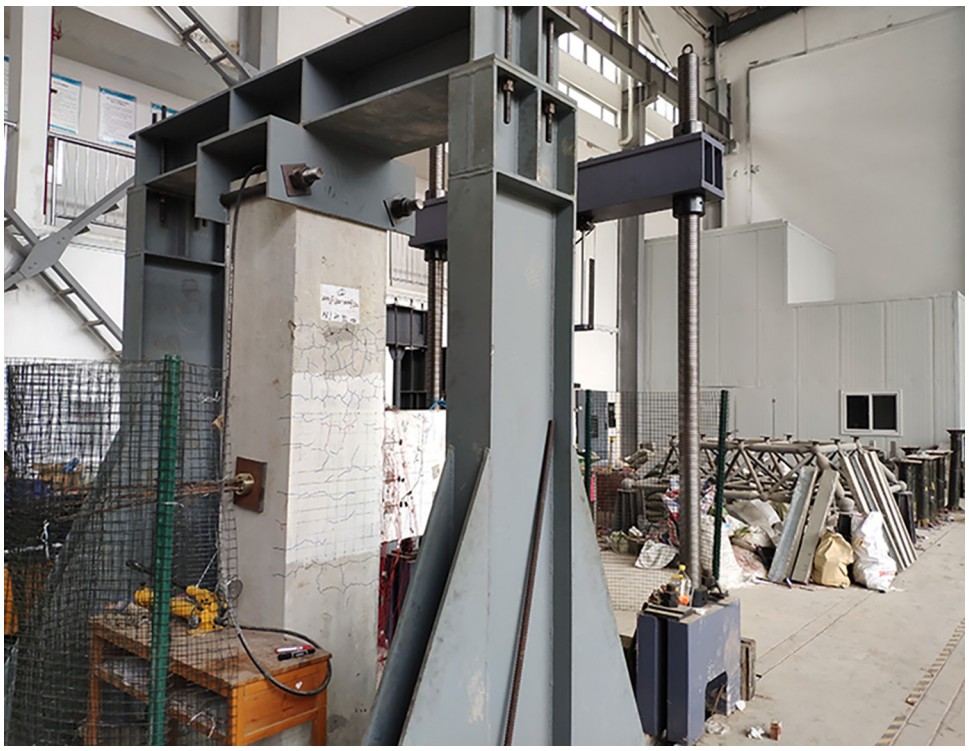

**Fig 8. Test loading (back).**

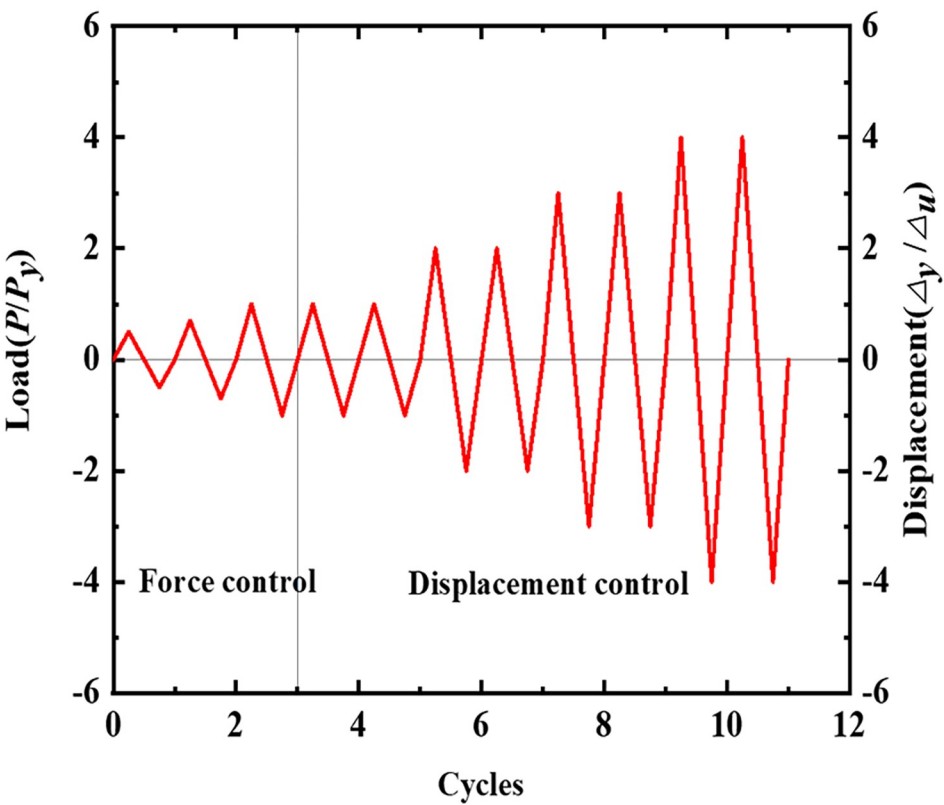

**Fig 9. Loading sequence.**

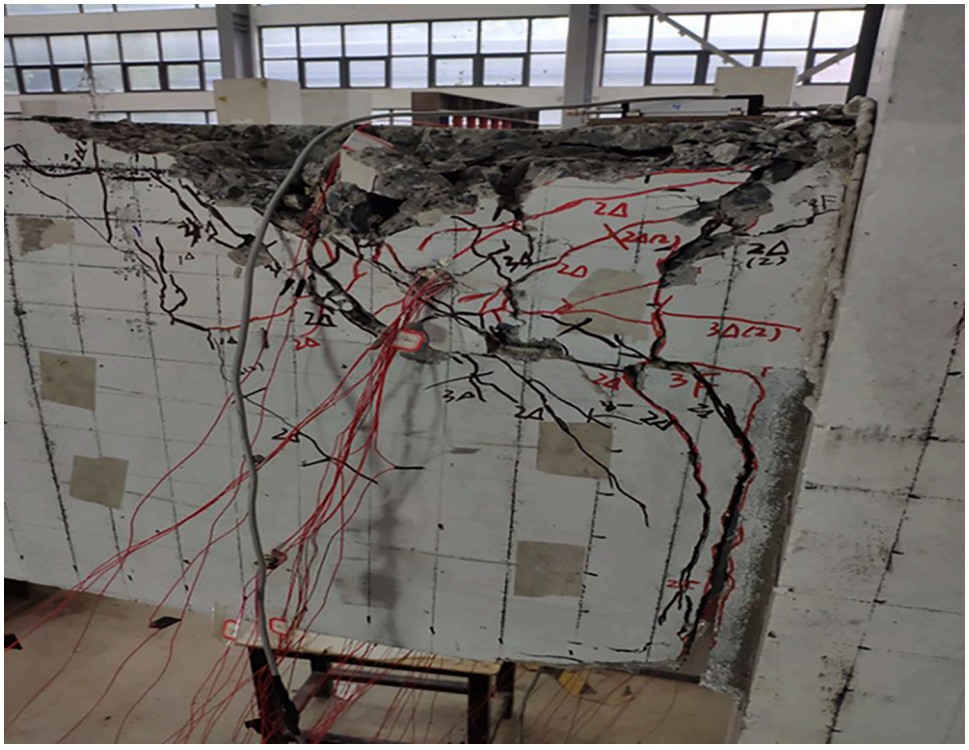

**Fig 10. Failure mode of SJ-1.**

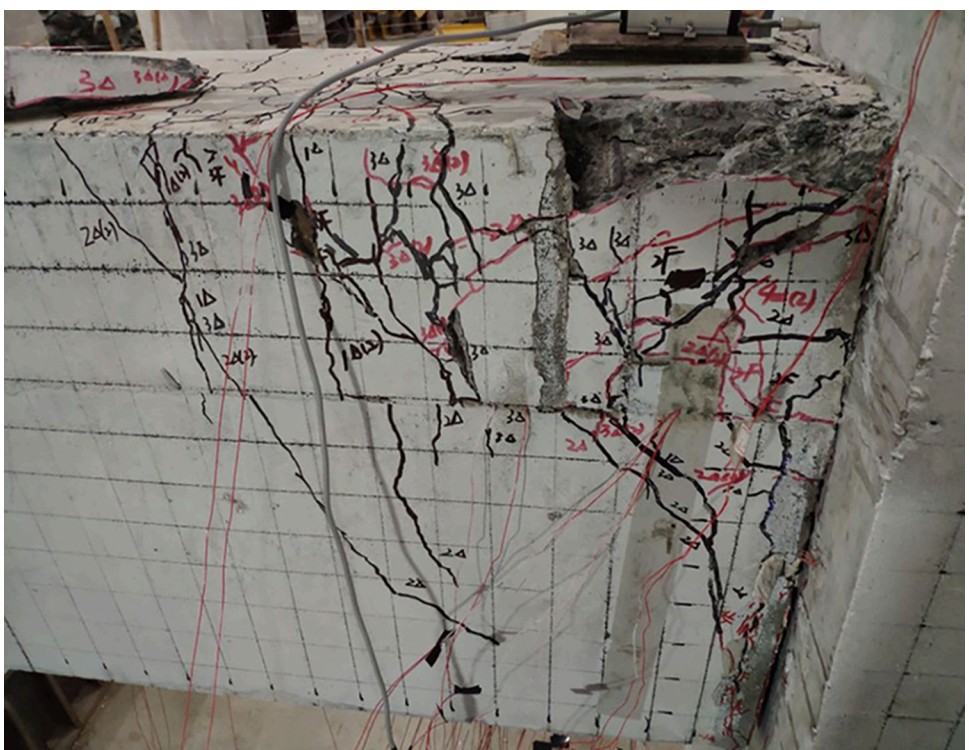

**Fig 11. Failure mode of SJ-2.**

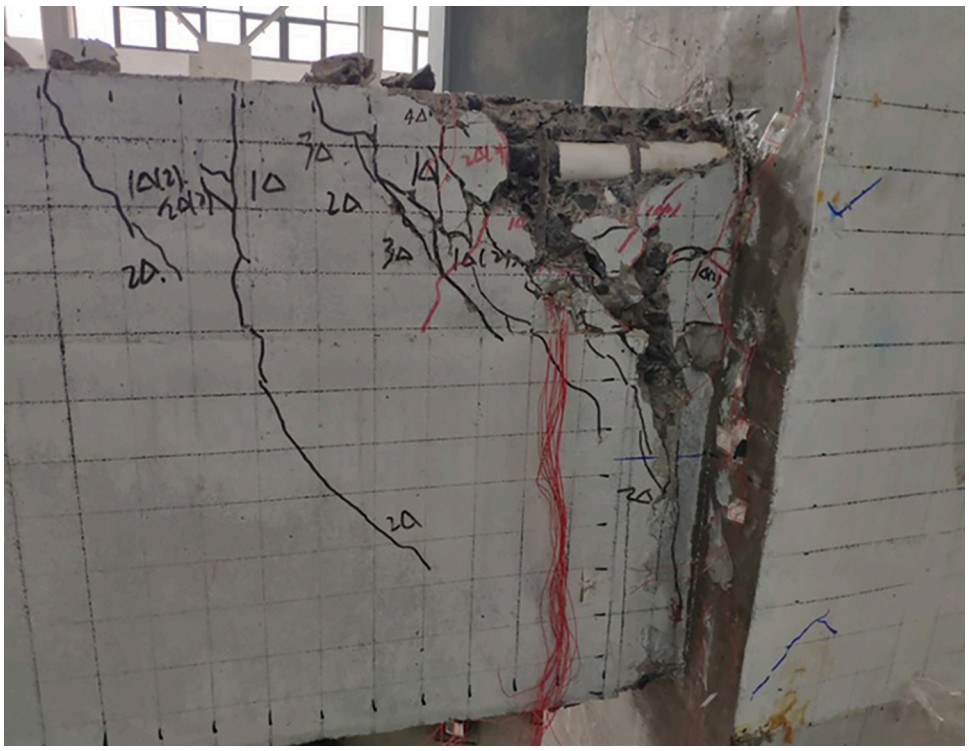

**Fig 12. Failure mode of SJ-3.**

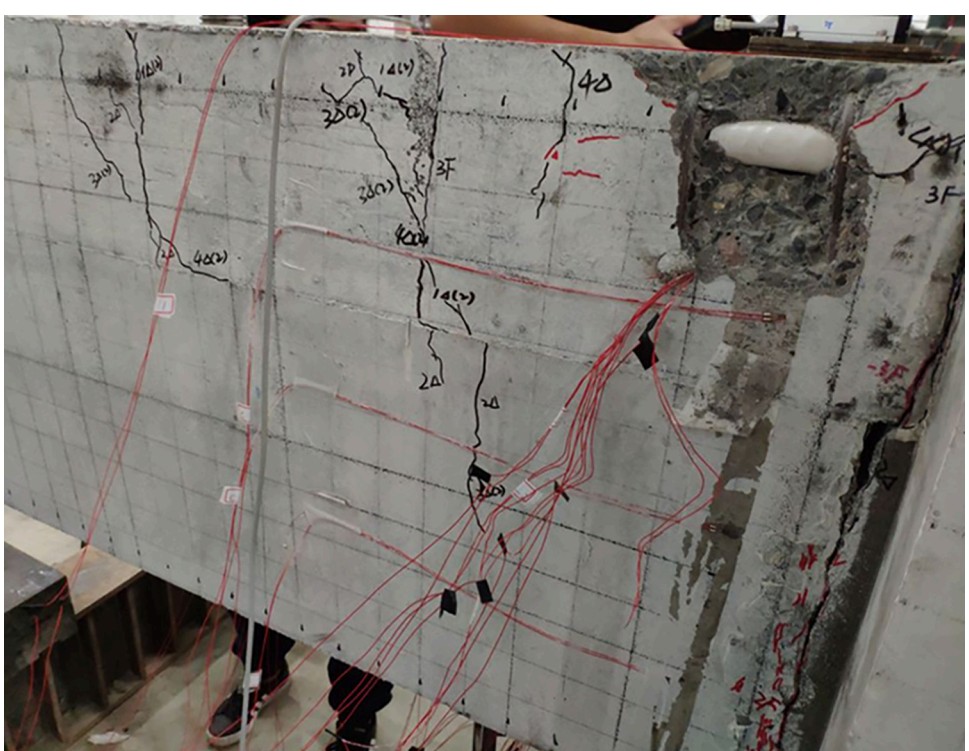

**Fig 13. Failure mode of SJ-4.**

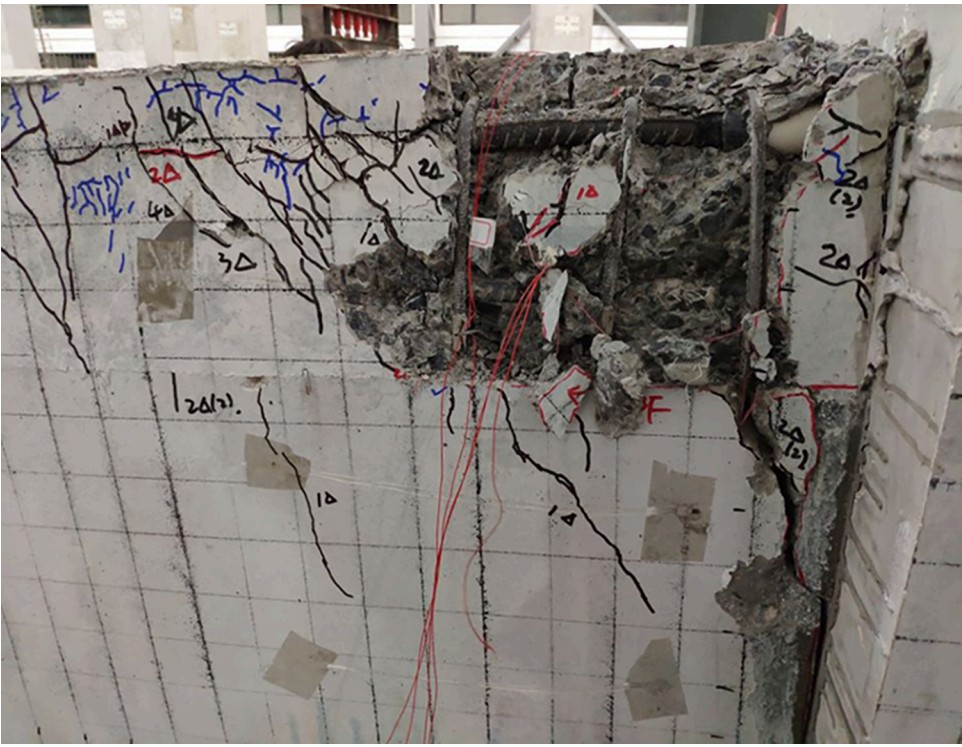

**Fig 14. Failure mode of SJ-5.**

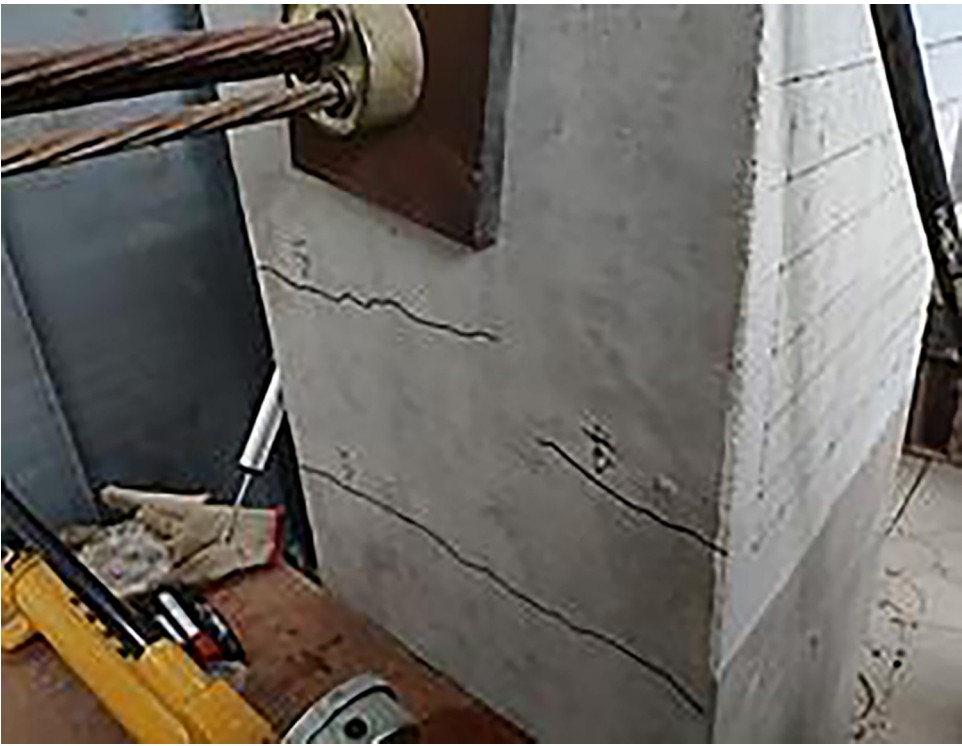

**Fig 15. Failure modes of SJ-1~SJ-5 columns.**

that in the lower part of the beam end because the energy dissipating bars and shear reinforcement in the upper part of the beam had to bear the tensile load. Increasing the reinforcement rate of the energy-consuming rebar led to an increase in the stiffness of the specimens and delayed the appearance of cracks. The damage to SJ-1, SJ-4, and SJ-5 indicated that the configuration of the shear reinforcement exacerbated the damage to the concrete at the fixed end of the beam.

## 4.2. Hysteresis curves and skeleton curves

The hysteresis curves of the five specimens are shown in Figs 16–21. The curves in the two loading directions are significantly different due to the asymmetrical reinforcement in the beam. As the beam end moved downward, the curve became fuller, and the energy consumption capacity improved. As the beam end moved upward, the curve became less full, and the energy dissipation capacity worsened. The pinching of the curve became more pronounced, indicating that the unbonded prestressed steel bar provided the specimen with improved recovery ability. Before the beam cracked, the curve indicated good linear elasticity and low energy consumption. As the test progressed, the number of cracks increased, and the cracks propagated. Thus, the curve became nonlinear, and the slope decreased. After the beam yielded, the stiffness of the specimen decreased as the plastic damage of the concrete accumulated. Under the same conditions, the strength of the high-energy-consuming steel bar reinforcement ratio specimen SJ-3 was higher than that of the low-energy-consuming steel bar reinforcement ratio specimens SJ-1 and SJ-2. The curve shrinkage of SJ-4 was more pronounced than that of SJ-1 and SJ-5, indicating that the shear reinforcement prevented the shrinkage of the specimen.

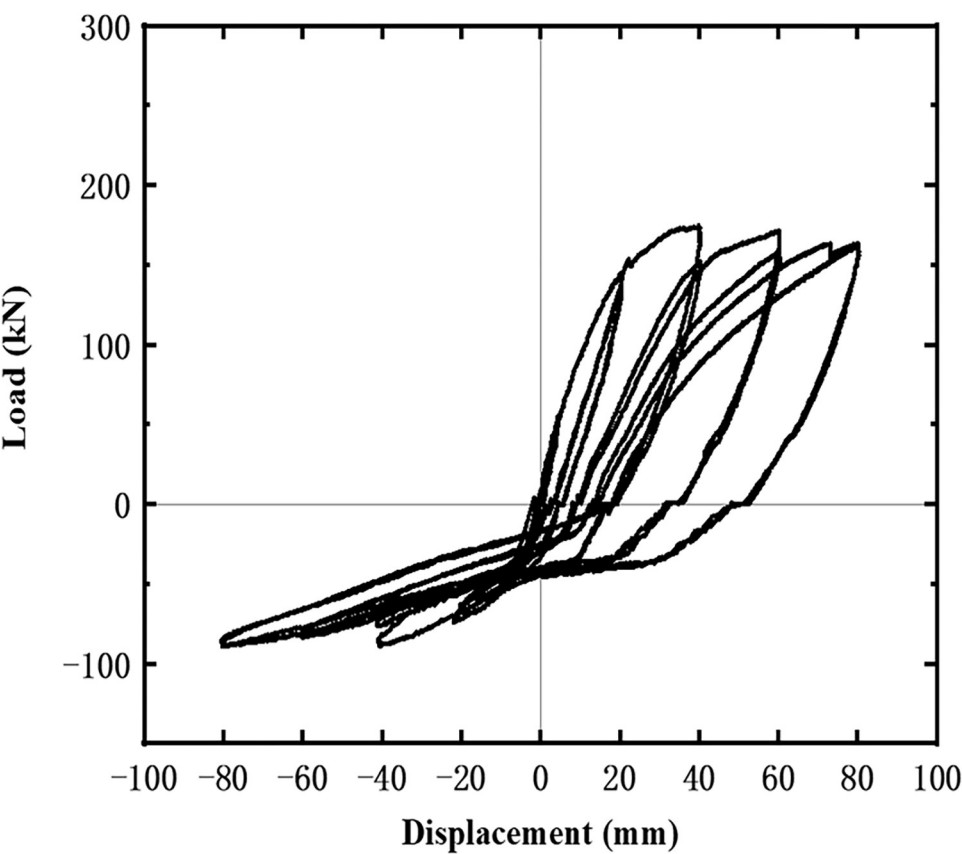

**Fig 16. Hysteresis curve of SJ-1.**

Fig 21 shows the skeleton curves of the five specimens. Since the energy dissipating bars and shear reinforcement are located in the upper part of the beam, the effective cross-sectional area increases during downward loading. Therefore, the load capacity is much higher for downward loading than for upward loading. The beam has good ductility in both loading directions. Due to the different reinforcement ratios of the energy dissipating rebar, the yielding stage of SJ-2 is longer than that of SJ-1 and SJ-3, indicating better ductility of SJ-2. Similarly, the ultimate load of SJ-1 is slightly higher than that of SJ-5, indicating that the shear reinforcement has a negligible effect on the bearing capacity of the specimens. Table 3 shows that the ratio of yield strength to ultimate strength is lower for downward loading than for upward loading. In addition, the ratio of yield strength to ultimate strength for upward loading decreases as the reinforcement ratio of the shear rebar increases.

### 4.3. Stiffness degradation

The stiffness degradation is expressed by the cut-line stiffness $K_i$, which is calculated as [34]:

$$K_i = (|+F_i| + |-F_i|)/(|+\Delta_i| + |-\Delta_i|) \tag{1}$$

where $+F_i$ and $-F_i$ are the peak loads for upward and downward loading, and $+\Delta_i$ and $-\Delta_i$ are the corresponding displacements. The stiffness degradation curves of the five specimens are shown in Fig 22. The results show that the stiffness of the specimens gradually decreases with larger displacements because of the accumulation of plastic damage to the concrete in the specimens. The higher initial stiffness of SJ-1 and SJ-3 compared to SJ-2 indicates that the initial

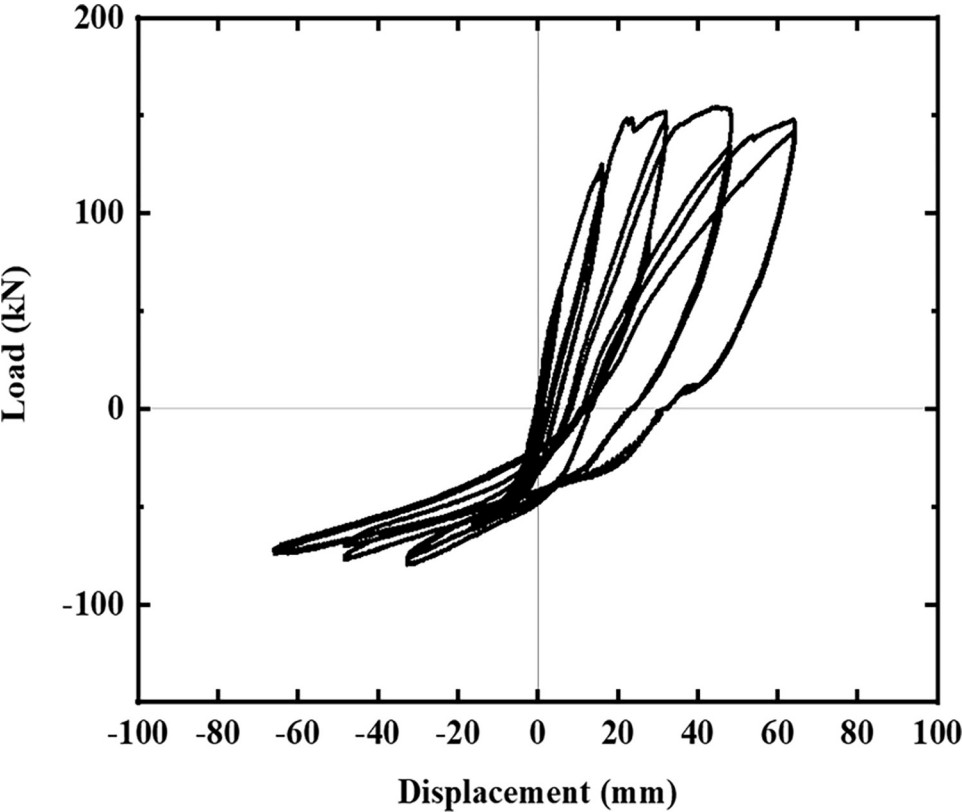

**Fig 17. Hysteresis curve of SJ-2.**

stiffness of the specimens increases with an increase in the reinforcement rate of the energy dissipating rebar. Similarly, an increase in the reinforcement rate of the shear rebar also increases the initial stiffness of the specimens. During the load control phase, the rate of stiffness degradation of the specimen is faster. After the specimens yielded, the rate of stiffness degradation decreased with an increase in the displacement. Comparing SJ-1 and SJ-2 shows that the stiffness degradation trend of SJ-3 was gentler, indicating that the higher the reinforcement rate of the energy-consuming rebar, the lower the rate of stiffness degradation of the specimens is. In addition, the correlation between the shear rebar reinforcement rate and the rate of stiffness degradation was not significant.

## 4.4. Ductility

The displacement ductility factor μ is used to express the ductility of the joints [37]:

$$\mu = \Delta_u/\Delta_y \tag{2}$$

where $\Delta_u$ is the displacement of the specimen when the load reached the maximum value and decreased to 85%, and $\Delta_y$ is the displacement when the specimen is loaded to the yield load. The ductility of the specimens is listed in Table 4. The ductility coefficients of the specimens loaded in the downward direction are between 2 and 4, which is similar to that of ordinary reinforced concrete joints. The ductility coefficient of the specimens under upward loading is between 13 and 21, indicating that the specimen had higher ductility when loaded upward. A comparison of the results of SJ-1, SJ-2, and SJ-3 indicated that the ductility of the specimen

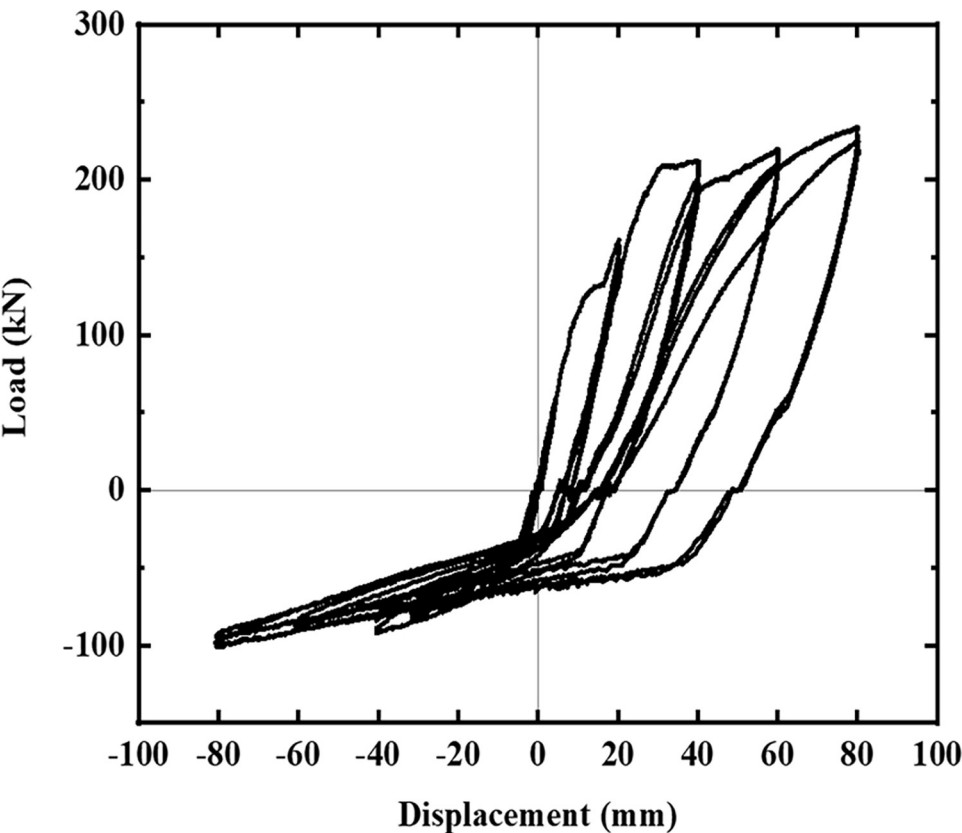

**Fig 18. Hysteresis curve of SJ-3.**

increased with a decrease in the reinforcement ratio of the energy-consuming steel bars. SJ-1 and SJ-2 have smaller ductility increments. In particular, the largest difference in the ductility was observed between SJ-1 and SJ-3, with a change of 5.31. Comparing SJ-1, SJ-4, and SJ-5 shows that the shear rebar reinforcement rate had less effect to the ductility of the specimens.

### 4.5. Energy dissipation

We use the cumulative energy dissipation and energy dissipation coefficient *E* to evaluate the energy dissipation capacity of the frame structure. [34]:

$$E = S_{ABCDA}/(S_{\Delta OBF} + S_{\Delta ODE}) \tag{3}$$

where $S_{ABCDA}$ is the area of the hysteresis loop ABCDA. $S_{\Delta OBF}$ and $S_{\Delta ODE}$ are the areas of the triangles OBF and ODE, respectively, as shown in Fig 23. The cumulative energy dissipation of the specimen increases with an increase in the displacement action, as shown in Fig 24. The curves of SJ-1, SJ-2, and SJ-3 demonstrate that the cumulative energy dissipation increases as the reinforcement ratio of the energy-consuming steel bars increases. A comparison of SJ-1, SJ-4, and SJ-5 shows that the effect of shear rebar reinforcement rate on the cumulative energy dissipation is small. Fig 25 shows an overall increasing trend of energy dissipation coefficient for the five specimens. However, the change of the curve sometimes shows a decreasing segment, which is because the compression of the prestressed reduces the plastic damage of the specimens.

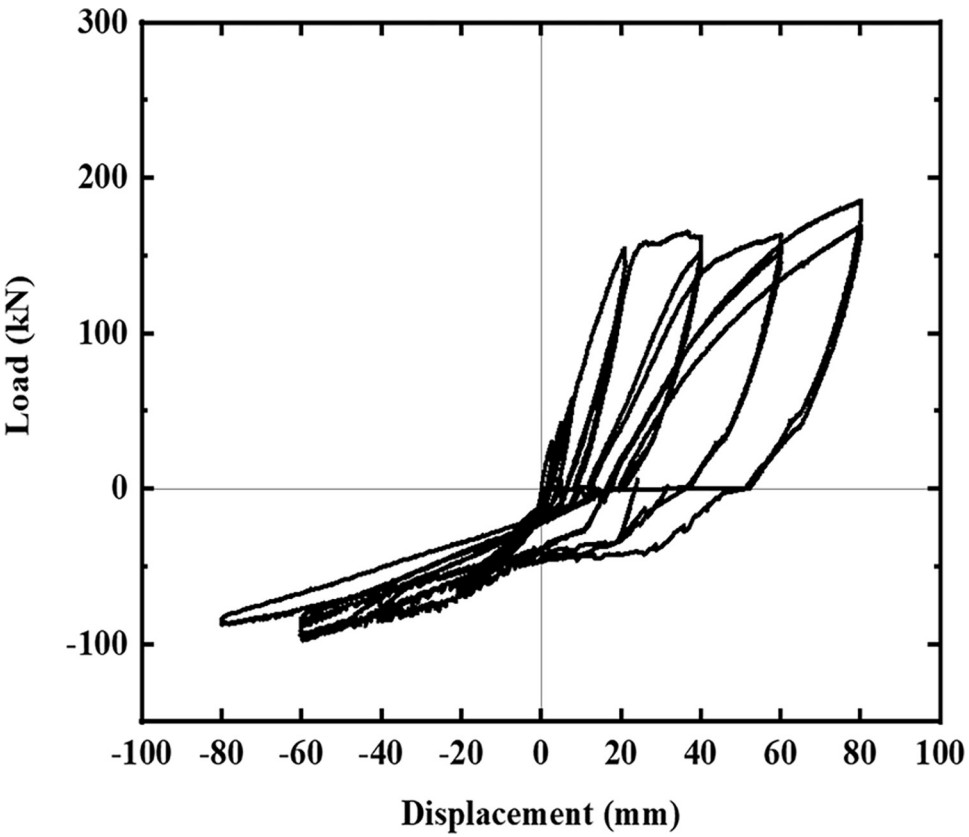

**Fig 19. Hysteresis curve of SJ-4.**

## 5. Bending moment calculation

The specimen beam is a laminated beam. Due to construction and test condition limitations, the composite layer has good bonding and coordinate deformation. High-strength mortar is used as the grouting material for the connection between the beams and columns. When the joints are deformed by forces, the mortar layer is susceptible to brittle damage when subjected to complex loads (extrusion, shear, impact, etc.), making it difficult to refine the analysis. The effect of high-strength mortar on the internal force of the beam-column interface is ignored to simplify the force analysis of the beam-column interface in this joint, and the focus is on the force condition of the fixed end section of the beam. The asymmetric reinforcement in the beam results in different force conditions of the specimens subjected to upward and downward loads. The upward loading and downward loading force conditions are analyzed separately.

### 5.1. Downward movement of the free end of the beam

When the free end of the beam moves downward, the upper part of the concrete at the fixed end of the beam is separated from the column. Since the anchorage length of the shear reinforcement is 15 d and the anchorage in the column is straight, the contribution of the shear reinforcement to the nodal bending moment after yielding is not considered. The mechanical model is shown in Fig 26.

**5.1.1. Plastic hinge length.** The deformation of each section in the plastic hinge region is equal to the curvature of the maximally deformed section. We use the equation proposed by

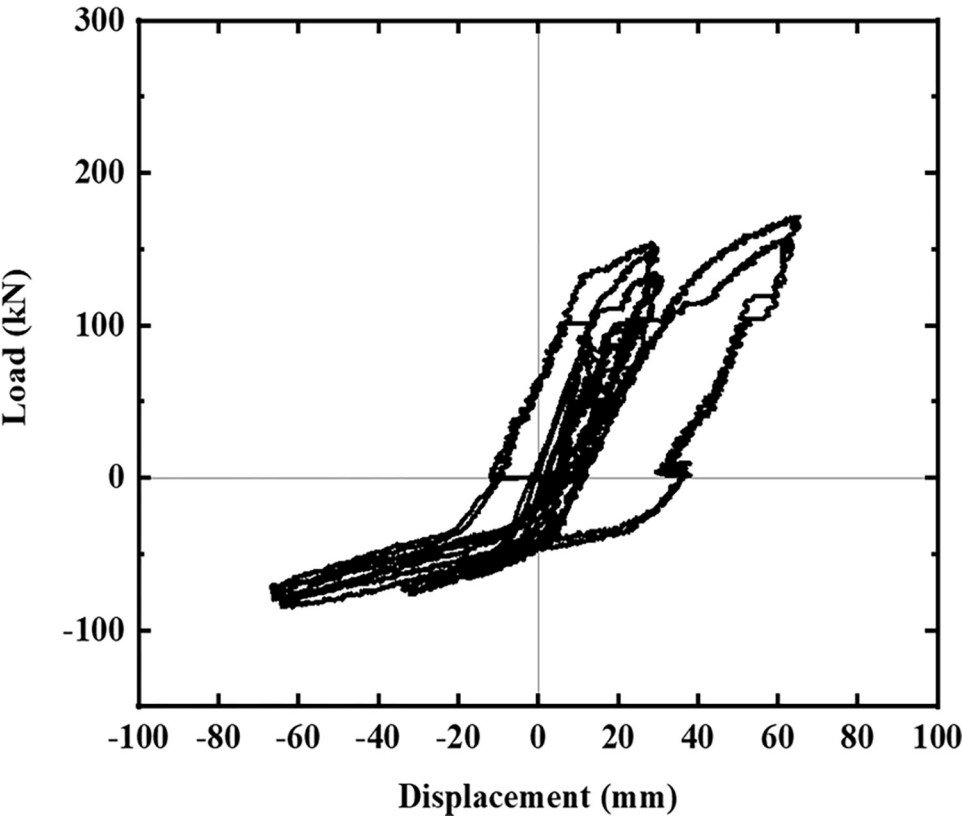

**Fig 20. Hysteresis curve of SJ-5.**

Paulay [34]:

$$L_P = 0.08L_s + 0.022f_y d_b \tag{4}$$

where $L_p$ is the plastic hinge length of the beam when loaded downward, $L_s$ is the shear span ratio of the member, $f_y$ is the design value of the tensile steel strength, and $d_b$ is the diameter of the tensile bar.

**5.1.2. Tensile strain of the energy-dissipating reinforcement.** When loaded downward, the energy-consuming reinforcement is deformed in tension, and the deformation increment is expressed as:

$$\varepsilon_e = \frac{\delta_e}{L_u} = \frac{(h - a_s' - x_c)\theta}{L_u} \tag{5}$$

where $\varepsilon_e$ is the tensile strain of the energy-dissipating steel bar, $\delta_e$ is the deformation length of the energy-consuming steel bar, $L_u$ is the length of the unbonded segment, $h$ is the height of the beam section, $a_s'$ is the distance between the resultant force point of the energy-dissipating reinforcement and the tensile edge of the beam section, $x_c$ is the height of the neutralization axis, and $\theta$ is the angle of the beam-column interface.

**5.1.3. Tensile strain of shear reinforcement.** The tensile strain of the shear steel bar in the plastic hinge area of the beam due to tensile deformation is calculated as follows:

$$\varepsilon_c = \frac{\delta_c}{L_p} = \frac{(h - a_s' - h_s - x_c)\theta}{0.08L_s + 0.022f_y d_b} \tag{6}$$

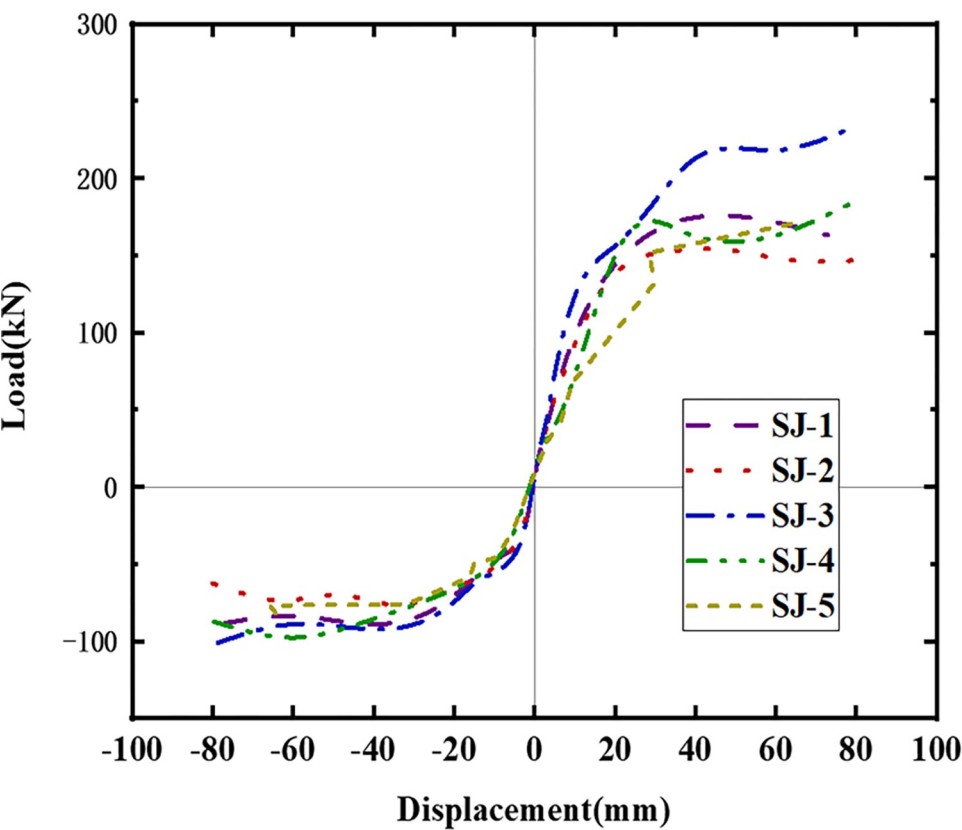

**Fig 21. Skeleton curves of five specimens.**

where $\varepsilon_c$ is the tensile strain of the shearing steel bar, $\delta_c$ is the tensile deformation length of the shearing steel bar, and $h_s$ is the distance from the point of the shearing steel bar to the point of the energy-dissipating steel bar.

**5.1.4. Tensile strain of prestressed reinforcement.**   The unbonded prestressed tensile strain is calculated as follows:

$$\varepsilon_p = \frac{\delta_p}{L + b_c} = \frac{(h/2 - x_c)\theta}{L + b_c} \tag{7}$$

**Table 3. Strength of five specimens in two loading directions.**

| Specimen | Loading Direction | Yield Load $f_y$ (kN) | Ultimate Load $f_u$ (kN) | Failure Load $f_l$ (kN) | $f_u/f_y$ |
|---|---|---|---|---|---|
| SJ-1 | up | 37.22 | 87.35 | 83.66 | 2.35 |
|  | down | 160.75 | 168.89 | 163.01 | 1.05 |
| SJ-2 | up | 33.75 | 76.81 | 73.27 | 2.28 |
|  | down | 141.86 | 154.02 | 147.04 | 1.09 |
| SJ-3 | up | 44.95 | 100.11 | 89.07 | 2.23 |
|  | down | 206.4 | 219.54 | 233.91 | 1.06 |
| SJ-4 | up | 25.78 | 87.39 | 87.39 | 3.39 |
|  | down | 156.23 | 185.35 | 163.08 | 1.19 |
| SJ-5 | up | 43.46 | 78.88 | 76.15 | 1.82 |
|  | down | 131.73 | 171.08 | 151.63 | 1.30 |

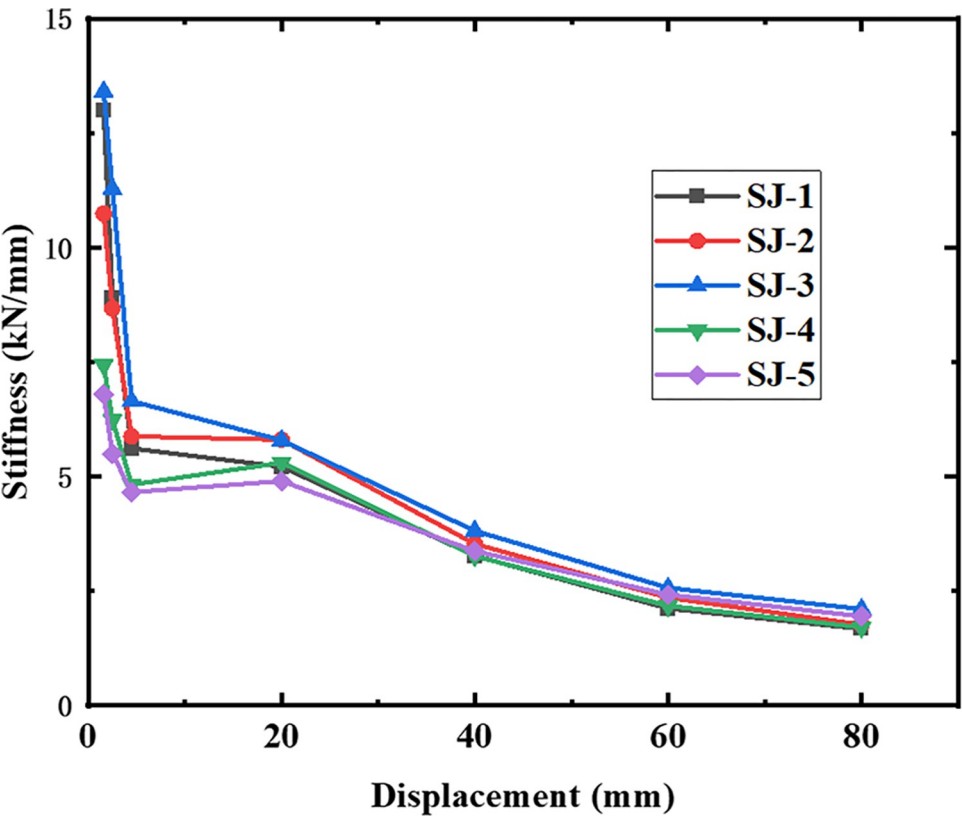

**Fig 22. Stiffness degradation of the five specimens.**

where $\varepsilon_p$ is the tensile strain of the prestressed steel bar, $\delta_p$ is the deformation length of the unbonded prestressed steel bar under positive action, $L$ is the beam length, and $b_c$ is the column width.

**5.1.5. Concrete strain at the compressive edge.** The concrete strain at the compressive edge is calculated as follows:

$$\varepsilon_{cs} = \frac{\delta_{cs}}{L_P} = \frac{x_c \cdot \theta}{0.08L_s + 0.022f_yd_b} \tag{8}$$

**Table 4. Ductility of the five specimens.**

| Specimen | Loading Direction | $\Delta_y$ (mm) | $\Delta_u$ (mm) | $\mu$ |
|---|---|---|---|---|
| SJ-1 | up | 4.32 | 80.01 | 18.52 |
| | down | 27.52 | 79.04 | 2.87 |
| SJ-2 | up | 3.13 | 64.11 | 20.48 |
| | down | 24.02 | 62.97 | 2.62 |
| SJ-3 | up | 5.83 | 77.03 | 13.21 |
| | down | 30.47 | 80 | 2.63 |
| SJ-4 | up | 4.72 | 79.6 | 16.86 |
| | down | 25.12 | 80.08 | 3.19 |
| SJ-5 | up | 4.48 | 66 | 14.73 |
| | down | 29.56 | 62.99 | 2.13 |

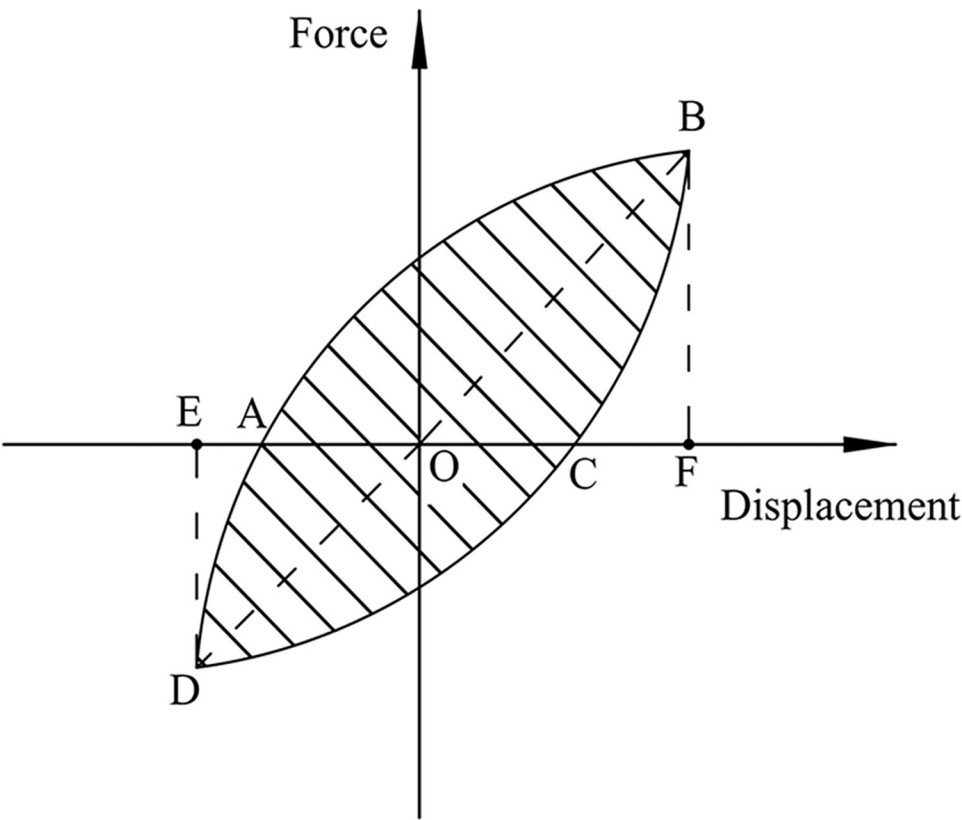

**Fig 23. Hysteresis loop.**

where $\varepsilon_{cs}$ is the concrete strain at the compressive edge of the fixed end section of the beam, and $\delta_{cs}$ is the compressive deformation of concrete.

**5.1.6. Compressive strain of the longitudinal bars at the bottom of the beam.**

$$\varepsilon_s = \frac{\delta_s}{L_p} = \frac{(x_c - a_s)\theta}{0.08L_s + 0.022f_yd_b} \tag{9}$$

where $\varepsilon_s$ is the compressive strain of the longitudinal bar at the bottom of the beam, $\delta_S$ is the deformation of the longitudinal bar at the bottom of the beam, $a_s$ is the distance between the resultant longitudinal reinforcement point at the bottom of the beam and the compressive edge of the section.

## 5.2. Upward movement of the free end of the beam

When the free end of the beam moves upward, the lower part of the concrete of the fixed end section of the beam is separated from the surface of the column, and the longitudinal reinforcement at the bottom of the beam does not participate in the force. Thus, the force on the contact surface of the beam and column is provided by four parts: the energy dissipating reinforcement, shear reinforcement, unbonded prestressed reinforcement, and compressed concrete. The mechanical model is shown in Fig 27.

**5.2.1. Plastic hinge length.** The stress state when the free end of the beam moves upward indicates that the reinforcement in the lower part of the beam does not reach the column and does not participate in the force. However, the upper part of the beam has energy dissipating

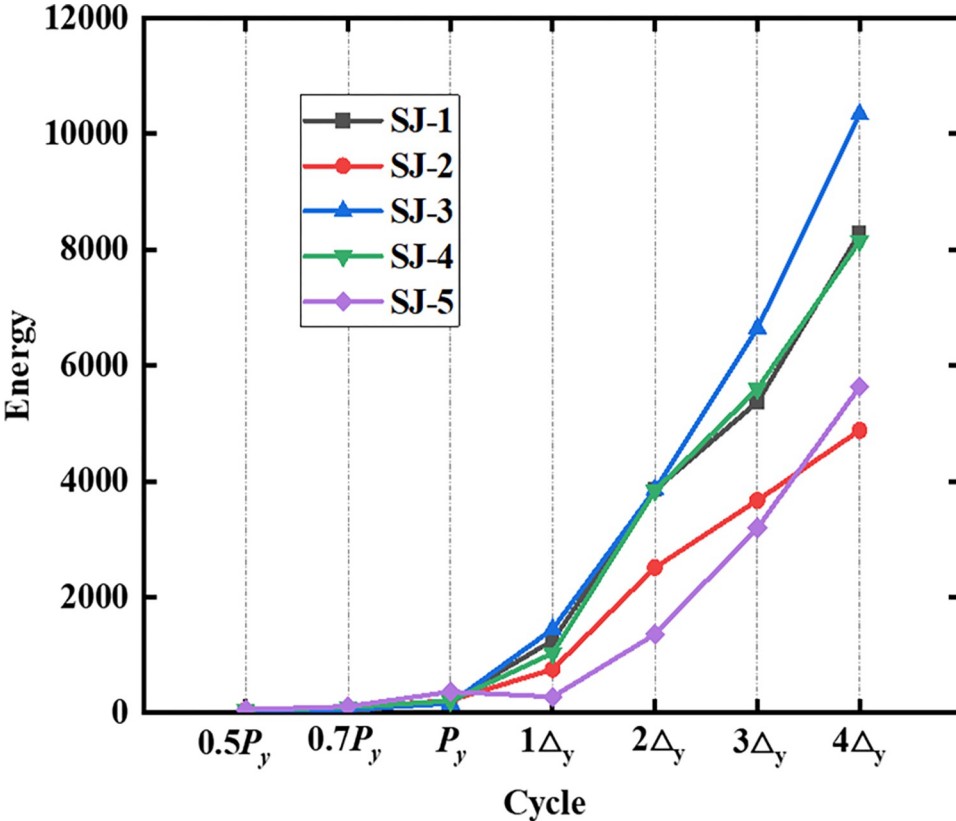

**Fig 24. Cumulative energy dissipation of the five specimens.**

reinforcement, shear reinforcement, and prestressing reinforcement participating in the force, resulting in deformation. The plastic hinge equation does not consider a change in the cross-sectional area, and Eq (4) can be used here.

**5.2.2. Compressive strain of the energy-dissipating reinforcement.**   When the free end of the beam moves upward, the deformation is concentrated in the unbonded section of the energy-dissipating reinforcement. Therefore, the deformation of the energy-dissipating steel bar is calculated as:

$$\varepsilon'_e = \frac{\delta'_e}{L_u} = \frac{(x_c - a'_s)\theta'}{L_u} \tag{10}$$

where $\varepsilon'_e$ is the compressive strain of the energy-dissipating steel bar, $\delta'_e$ is the compressive deformation of the energy-dissipating steel bar, and $\theta'$ is the beam-column interface rotation angle.

**5.2.3. Compressive strain of shear reinforcement.**

$$\varepsilon'_c = \frac{\delta'_c}{L'_P} = \frac{(x_c - a'_s - h_s)\theta'}{0.08L_s + 0.022f_yd_b} \tag{11}$$

where $\varepsilon'_c$ is the compressive strain of the shear reinforcement and $\delta'_c$ is the compressive deformation length of the shear reinforcement.

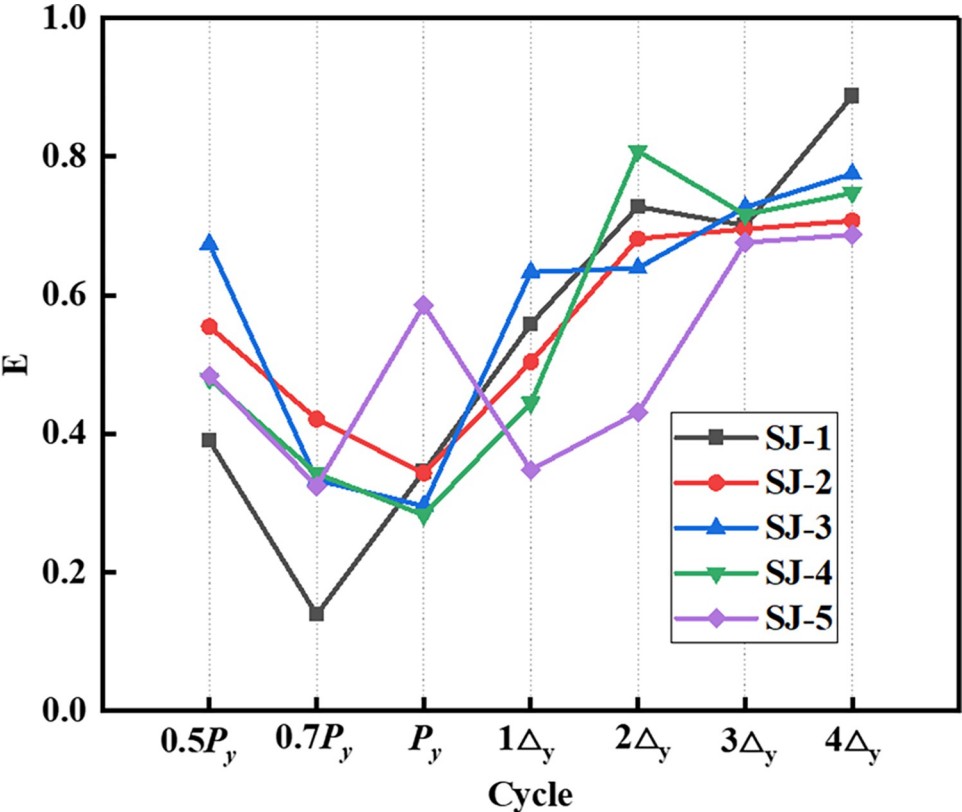

**Fig 25. Energy dissipation coefficient of the five specimens.**

**5.2.4. Tensile strain of prestressed reinforcement.** The unbonded prestressed tensile strain is calculated as follows:

$$\varepsilon'_p = \frac{\delta'_p}{L + b_c} = \frac{(h/2 - x_c)\theta'}{L + b_c} \tag{12}$$

where $\varepsilon'_p$ is the tensile strain of the prestressed steel bar, and $\delta'_p$ is the tensile strain increment of the prestressed reinforcement.

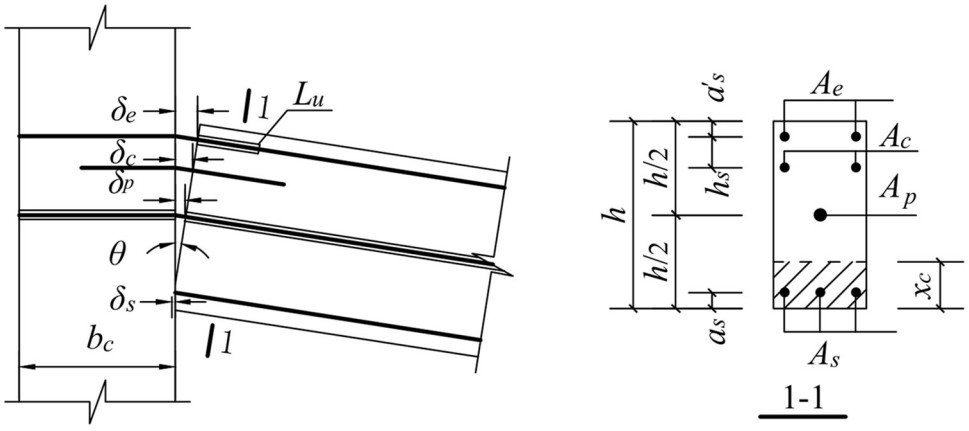

**Fig 26. Mechanical model for downward load.**

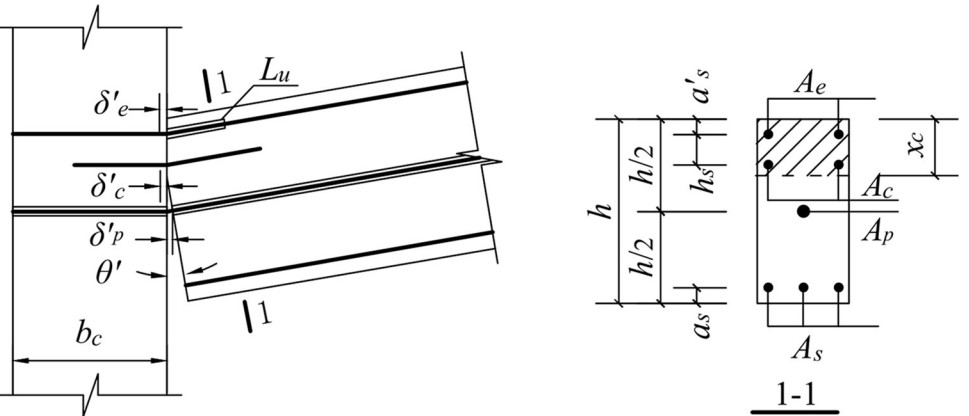

**Fig 27. Mechanical model for upward load.**

**5.2.5. Concrete strain at the compressive edge.** The concrete strain at the compressive edge is calculated as:

$$\varepsilon'_{cs} = \frac{\delta'_{cs}}{L'_p} = \frac{x_c \cdot \theta'}{L_p} \tag{13}$$

We substitute Eq (4) into Eq (13):

$$\varepsilon'_{cs} = \frac{x_c \cdot \theta'}{0.08L_s + 0.022f_yd_b} \tag{14}$$

where $\varepsilon'_{cs}$ is the concrete strain at the compressive edge of the beam-column interface, and $\delta'_{cs}$ is the compressive deformation of concrete.

## 5.3. Internal force analysis

**5.3.1. Material constitutive relations.** We use the constitutive model in the Standard GB50010-2010 to calculate the stress-strain relationship between the concrete and steel bars [30]. Therefore, the stress-strain behavior of the dissipative and prestressed steel bars is described by a double slash constitutive model. The constitutive model of concrete under uni-axial stress is used to describe the stress-strain behavior of concrete under compression.

The stresses resulting from the strain of each part of the reinforcement and compressed concrete on the beam-column interface are calculated using a principal structure model of the reinforcement and concrete to obtain their internal forces. It is found that the internal forces are expressions of $\theta$, $\theta'$, and $x_c$.

Downward load:

$$\begin{cases} T_e = f(\theta, x_c) \\ T_c = f(\theta, x_c) \\ T_p = f(\theta, x_c) \\ C = f(\theta, x_c) \\ C_S = f(\theta, x_c) \end{cases} \tag{15}$$

Upward load:

$$\begin{cases} C_e = f(\theta', x_c) \\ C_c = f(\theta', x_c) \\ T_p = f(\theta', x_c) \\ C = f(\theta', x_c) \end{cases} \tag{16}$$

where $T_e$ and $C_e$ are the resultant force of the energy-dissipating steel bar under tension and compression, respectively, $T_c$ and $C_c$ are the resultant force of the shear reinforcement under tension and compression, respectively. $T_p$ is the resultant force of the prestressed reinforcement under tension, $C$ is the resultant force of the concrete under compression, $C_s$ is the resultant force of the longitudinal reinforcement at the bottom of the beam under compression.

**5.3.2. Balance equation.** Fig 28 shows a schematic of the calculation of the normal section at the fixed end of the beam. The internal forces on the beam-column interface are provided by the energy-dissipating steel bars, shear reinforcement, prestressed reinforcement, longitudinal rebar at the bottom of the beam, and compressed concrete. Thus, the mechanical equilibrium equations for the free end of the beam in two motion directions are as follows.

$$T_e + T_c + T_p + C + C_s = 0 \tag{17}$$

$$C_e + C_c + T_p + C = 0 \tag{18}$$

**5.3.3. Bending moment equation.** In this paper, the moment is the geometric center of the fixed end section of the beam (the point of prestressed reinforcement cohesion), and the normal stress of the compressed concrete is calculated using an equivalent rectangle, resulting in the following formula for calculating the bending moment when the free end of the beam moves downward and upward, respectively:

$$M_1 = T_e\left(\frac{h}{2} - a'_s\right) + T_c\left(\frac{h}{2} - a'_s - h_s\right) + C \cdot \frac{h - x}{2} + C_s\left(\frac{h}{2} - a_s\right) \tag{19}$$

$$M_2 = C_e\left(\frac{h}{2} - a'_s\right) + C_c\left(\frac{h}{2} - a'_s - h_s\right) + C \cdot \frac{h - x}{2} \tag{20}$$

where $M_1$ and $M_2$ are the bending moments of the downward and upward movements of the

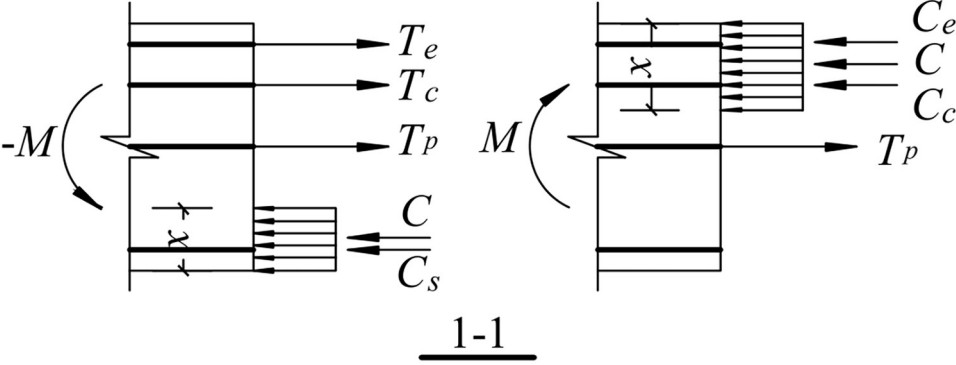

**Fig 28. Calculation diagram of the bending moment in the normal section.**

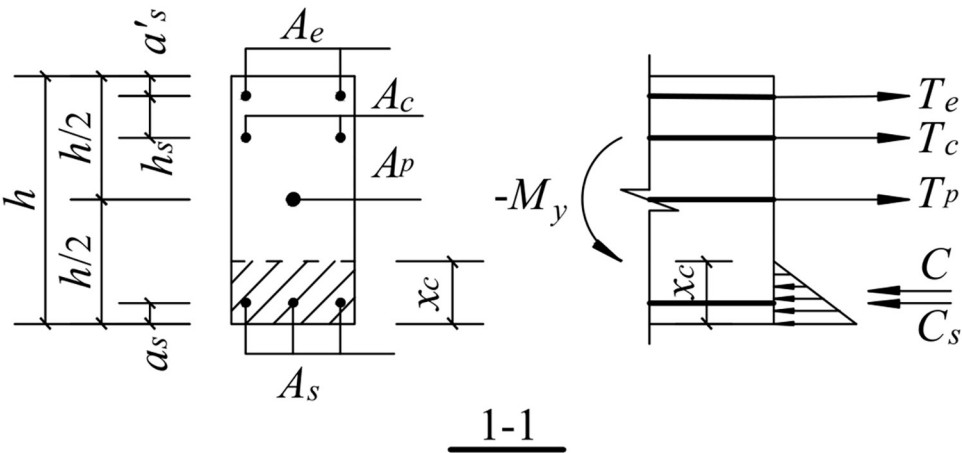

**Fig 29. Schematic diagram of the yield moment calculation for downward load.**

free end of the beam, respectively. $x$ is the equivalent compression height of the concrete; it is defined as $x = \beta_1 x_c$, where $\beta_1$ is 0.8 [30].

**5.3.4. The yield moment.** The joint structure shows that the yield moment of the joint is the cross-sectional moment when the energy-consuming reinforcement yields in compression or in tension, as shown in Figs 29 and 30. $A_e$ is the cross-sectional area of the energy-dissipating steel bar, $A_c$ is the cross-sectional area of the shear steel bar, $A_p$ is the cross-sectional area of the prestressed steel bar, and $A_s$ is the cross-sectional area of the longitudinal bar at the bottom of the beam.

The standard (GB50010-2010) gives $f_y = f'_y$; therefore, the internal force of the energy-dissipating reinforcement at yielding is $T_e = f_y A_e$ [30]. Since the unbonded section of the energy-dissipating steel bars is inside the beam, the buckling of the steel bars does not reach the expected strength under compression theoretically; thus, the flexion reduction coefficient is $\gamma = 0.5$, i.e., $C_e = \gamma f_y A_e$.

The downward movement of the free end of the beam is shown in Fig 29:

We obtain the following from the force equilibrium conditions:

$$C = f_y A_e + T_c + T_p - C_s \tag{21}$$

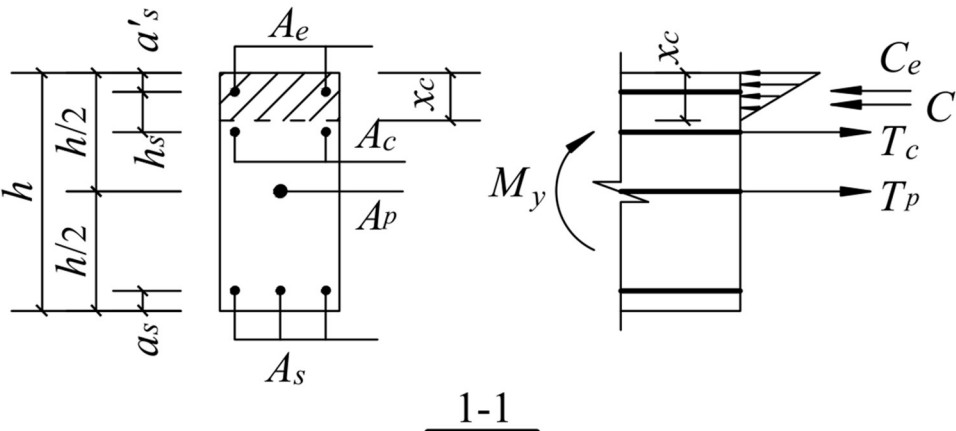

**Fig 30. Schematic diagram of the yield moment calculation for upward load.**

and the following from the moment equilibrium conditions:

$$-M_y = f_y A_e \left(\frac{h}{2} - a_s'\right) + T_c \left(\frac{h}{2} - a_s' - h_s\right) + C \cdot \left(\frac{h}{2} - \frac{x_c}{3}\right) + C_s \left(\frac{h}{2} - a_s\right) \tag{22}$$

The upward movement of the free end of the beam is shown in Fig 30:
We obtain the following from the force equilibrium conditions:

$$C = T_c + T_p - \gamma f_y' A_e \tag{23}$$

and the following from the moment equilibrium condition:

$$M_y = \gamma f_y' A_e \left(\frac{h}{2} - a_s'\right) - T_c \left(\frac{h}{2} - a_s' - h_s\right) + C \cdot \left(\frac{h}{2} - \frac{x_c}{3}\right) \tag{24}$$

where $-M_y$ and $M_y$ are the yield bending moments for the downward and upward movements of the free end of the beam, $A_e$ is the cross-sectional area of the energy-dissipating steel bar, $f_y$ and $f_y'$ are the design values of the tensile strength and compressive strength of the energy-dissipating steel bars, respectively.

**5.3.5. Ultimate bending moment.** In the ultimate state, the concrete protective layer at the beam-column interface has been crushed and peeled off. Therefore, the role of the concrete protective layer is not considered. The energy-consuming reinforcement has yielded; thus, its contribution is not considered. In addition, the contribution of the shear reinforcement to the ultimate bending moment is not considered because it has already yielded in the ultimate state for safety reasons. As shown in Figs 31 and 32, the normal stress distribution of the compression of the concrete shows a rectangular distribution. The concrete expression is: $C = \alpha_1 f_c (b - 2c)(x - c)$ where $b$ is the width of the beam section, $c$ is the thickness of the protective concrete layer, and $\alpha_1$ is 1.0.

The downward movement of the free end of the beam is shown in Fig 31:
The equilibrium conditions of the forces are:

$$C = T_e + T_p - C_s \tag{25}$$

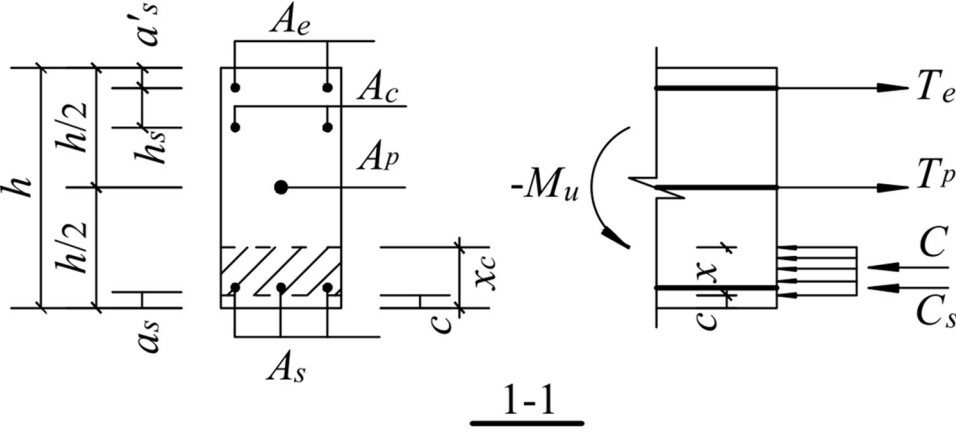

**Fig 31. Schematic diagram of the ultimate moment calculation for downward load.**

Therefore,

$$\alpha_1 f_c (b - 2c)(x - c) = f_y A_e + T_p - f_{yl} A_s \tag{26}$$

The moment equilibrium conditions are:

$$-M_u = f_y A_e \left( \frac{h}{2} - a'_s \right) + \alpha_1 f_c (b - 2c)(x - c) \cdot \frac{h - 2c - x}{2} + f_{yl} A_s \left( \frac{h}{2} - a_s \right) \tag{27}$$

The upward movement of the free end of the beam is shown in Fig 32:
The equilibrium conditions of the forces are:

$$C = T_p - C_e \tag{28}$$

Therefore,

$$\alpha_1 f_c (b - 2c)(x - c) = T_p - f'_y A_e \tag{29}$$

The moment equilibrium conditions are:

$$M_u = f'_y A_e \left( \frac{h}{2} - a'_s \right) + \alpha_1 f_c (b - 2c)(x - c) \cdot \frac{h - 2c - x}{2} \tag{30}$$

where $-M_u$ and $M_u$ are the limit bending moments for the downward and upward movements of the free end of the beam, and $f_{yl}$ is the design value of the compressive strength of the longitudinal reinforcement at the bottom of the beam.

## 5.4. Verification of calculation results

The comparison of flexural capacity calculated by the equation and obtained from the test is shown in Table 5. On the one hand, the maximum ratio of the calculated results to the test results is 1.26, and the maximum error is 26.5%. Since the strain detection of reinforcement and concrete is easily disturbed by peripheral factors, such as initial cracking of concrete and initial strain caused by the self-weight of the specimen. Therefore, the error between the equation calculation result of SJ-4 and the test result is large, and the calculation error of other specimens is less than 20%. On the other hand, the average error of the specimen samples (0–10) in the yielding stage is 8.07%, and the standard deviation is 7.95%. The mean error of the

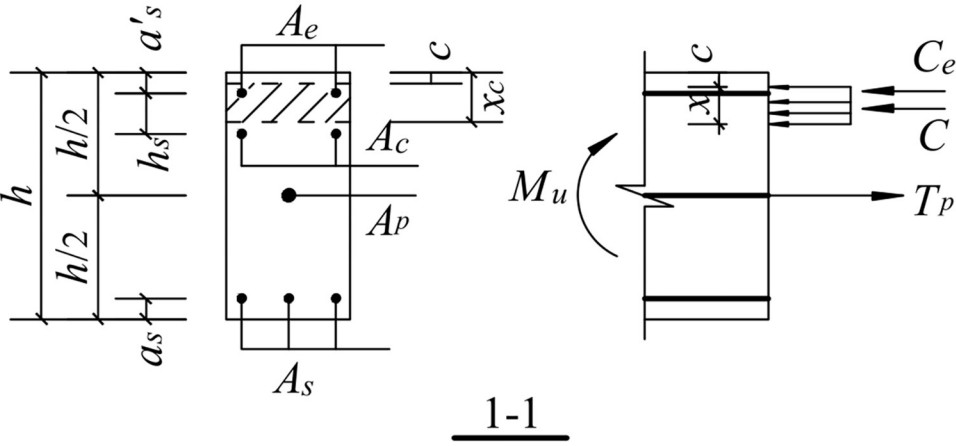

**Fig 32. Schematic diagram of the ultimate moment calculation for upward load.**

**Table 5. Comparison of calculated and test values.**

| Phase | Specimen | $M^c$ [a] (kN•m) | $M^t$ [b] (kN•m) | $M^c/M^t$ [c] | Error (%) |
|---|---|---|---|---|---|
| $-M_y$ | SJ-1 | 290.59 | 273.28 | 1.06 | 6.34% |
|  | SJ-2 | 275.54 | 241.17 | 1.14 | 14.25% |
|  | SJ-3 | 353.62 | 350.88 | 1.01 | 0.78% |
|  | SJ-4 | 271.13 | 265.59 | 1.02 | 2.08% |
|  | SJ-5 | 246.95 | 223.94 | 1.10 | 10.28% |
| $M_y$ | SJ-1 | 62.64 | 63.28 | 0.99 | -1.02% |
|  | SJ-2 | 55.58 | 57.38 | 0.97 | -3.13% |
|  | SJ-3 | 77.74 | 76.42 | 1.02 | 1.72% |
|  | SJ-4 | 55.44 | 43.83 | 1.26 | 26.50% |
|  | SJ-5 | 63.12 | 73.88 | 0.85 | -14.57% |
| $-M_u$ | SJ-1 | 259.04 | 287.12 | 0.90 | -9.78% |
|  | SJ-2 | 238.27 | 261.83 | 0.91 | -9.00% |
|  | SJ-3 | 300.81 | 373.22 | 0.81 | -19.40% |
|  | SJ-4 | 266.93 | 315.10 | 0.85 | -15.29% |
|  | SJ-5 | 264.70 | 290.84 | 0.91 | -8.99% |
| $M_u$ | SJ-1 | 133.49 | 148.50 | 0.90 | -10.11% |
|  | SJ-2 | 116.27 | 130.58 | 0.89 | -10.96% |
|  | SJ-3 | 169.60 | 170.19 | 1.00 | -0.35% |
|  | SJ-4 | 127.55 | 148.56 | 0.86 | -14.14% |
|  | SJ-5 | 127.55 | 134.10 | 0.95 | -4.88% |

[a] the calculated value.

[b] the test value.

[c] the ratio of the calculated bending moment to the bending moment obtained from the test.

specimen samples (11–20) in the limit phase is 10.29% with a standard deviation of 5.05%. Generally, the calculated results are in good agreement with the test results. Therefore, the proposed flexural capacity equation is reasonable.

## 6. Conclusion

Cycling loading tests were conductecd on PPEFF joints to analyze the damage modes, hysteresis curves, skeleton curves, stiffness degradation, ductility, and energy dissipation capacity of five specimens. In addition, the effects of different reinforcement rates of the energy dissipating bars and shear reinforcement on the seismic performance of the joints were compared. An equation of the flexural capacity of the joints was proposed. The effect of the shear reinforcement in different force stages was considered in deriving the flexural capacity equation of the PPEFF joints. A mechanical model of the beam-column interface of the joints was established to conduct a force analysis of the reinforcement and concrete. An internal force equilibrium equation and bending moment equilibrium equation were established, and the flexural capacity equations in the yielding and ultimate stages were determined. The findings of this paper are as follows.

1. The test results showed that the damage to all five specimens was beam bending damage. The damage near the plastic hinge region of the beam was more severe, which is consistent with the ideal joint damage mode.

2. Since the energy dissipating bars and shear reinforcement were located in the upper part of the beam, the joint bearing capacity and energy dissipation performance of the beam were higher when the free end moved downward rather than upward, but the ductility was lower.

3. Increasing the reinforcement rate of the energy-consuming and shear steel bars enhanced the energy dissipation capacity of the joint. In addition, decreasing the reinforcement rate of energy dissipating bars enhanced the ductility of the joint. The greatest change in the ductility was observed when the reinforcement rate of energy-consuming bars ranged from 0.38% to 0.59%, with 5.31.

4. A comparison of the calculated and experimental results showed that the average errors in the yielding stage and ultimate stage were 8.07% and 10.29%, respectively, and the standard deviations were small. The good agreement verifies the suitability of the proposed equation, which can be used for the theoretical study of PPEFF joints.

5. The post-earthquake bearing capacity of the PPEFF joints will be analyzed in-depth in a future study.

## Supporting information

**S1 File. Relevant data underlying the findings described in manuscript.**
(XLSX)

## Acknowledgments

The authors would like to thank all experts who provided data. Also, sincerely thanks to the reviewers for their useful comments on this paper.

## Author Contributions

**Conceptualization:** Yunlin Jiang.

**Data curation:** Yunlin Jiang.

**Formal analysis:** Jie Cai, Yunlin Jiang.

**Funding acquisition:** Shuyang Li.

**Methodology:** Jie Cai, Yunlin Jiang.

**Project administration:** Jie Cai, Shuyang Li.

**Resources:** Jie Cai, Shuyang Li.

**Software:** Yunlin Jiang.

**Supervision:** Jie Cai.

**Validation:** Jie Cai.

**Visualization:** Jie Cai, Yunlin Jiang.

**Writing – original draft:** Yunlin Jiang.

**Writing – review & editing:** Jie Cai, Yunlin Jiang.

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
