## [Decision Letter · Decision Letter 0]

14 Mar 2022

PONE-D-22-04046The Flexural Mechanical Properties of Prestressed Efficiently Prefabricated Beam-Column ConnectionsPLOS ONE

Dear Dr. Jiang,

Thank you for submitting your manuscript to PLOS ONE. After careful consideration, we feel that it has merit but does not fully meet PLOS ONE’s publication criteria as it currently stands. Therefore, we invite you to submit a revised version of the manuscript that addresses the points raised during the review process. Please, address all the comments made by the reviewers. More details on the design are required and the presentation (abstract, literature survey, theoretical part and conclusions) should be improved. 

We look forward to receiving your revised manuscript.

Kind regards,

Antonio Riveiro Rodríguez, PhD

Academic Editor

PLOS ONE

[This research was financially supported by Science and Technology for the Economy 2020 (2020ZLSH08).]

 [No]

[NO authors have competing interests]. 

Reviewers' comments:

Reviewer's Responses to Questions

**Comments to the Author**

1. Is the manuscript technically sound, and do the data support the conclusions?

Reviewer #1: Partly

Reviewer #2: Partly

Reviewer #3: No

2. Has the statistical analysis been performed appropriately and rigorously? 

Reviewer #1: N/A

Reviewer #2: Yes

Reviewer #3: Yes

3. Have the authors made all data underlying the findings in their manuscript fully available?

Reviewer #1: Yes

Reviewer #2: Yes

Reviewer #3: No

4. Is the manuscript presented in an intelligible fashion and written in standard English?

Reviewer #1: Yes

Reviewer #2: No

Reviewer #3: Yes

5. Review Comments to the Author

Reviewer #1: This manuscript presents an experimental program on precast prestressed efficiently fabricated beam column joints. The research is interesting and of value to practice application of precast structures. The authors are advised to address my following comments:

1. Section 1.1: please provide more information on the design of the proposed beam-column joints. For example, how the joint is fabricated. How the prestressing is applied. How the energy consuming steel bars are installed. Please also justify the advantages of the proposed joints compared with other forms by others.

2. It seems that the shear force of the beam is resisted only by the steel bars. Please provide more information on the design of linking bars at the beam-column interface. Whether the steel bars are sufficiently strong and stiff to resist the shear force?

3. Figure 5: the energy dissipation in the positive loading direction is much larger than in the negative direction, which is due to the use of energy consuming steel bars at the top region of the beam. Please justify such treatment in the text.

4. Section 3.4 is too simple. Please discuss in more detail on the comparison between the predicted results and the test results for the specimens.

Reviewer #2: This paper is technically sound, but language needs substantial improvement. This reviewer is happy to go through the technical parts after the language is improved. A professional language editing service may be sought.

Reviewer #3: 1-What do you mean flexural mechanical properties? Did you mean flexural behavior or strength?

2-Abstract: The main and important results should be presented.

3-Keywords:

-Abbreviation cannot be used.

-Please do not use the same words in title and keywords.

4-What is the novelty of this study?!

-The novelty should be presented in the end of paragraph of introduction.

-The aim of study should be presented at the first paragraph of methodology.

5-Did you apply the load as load-control? but why did not you use displacement control? How did you control the applied force of the jack? Did you consider the rate of loading from a test standard like ASTM?

6-The quality of figures, especially in the result, is poor!

7-How did you avoid the shear failure in your test setup?

8- The literature review can be more comprehensive for example following works on experiments and simulations of reinforced concrete (RC) element could be cited.

*2020. Performance of fixed beam without interacting bars. Frontiers of Structural and Civil Engineering, 14(5), pp.1180-1195.

*2019. Performance of a novel bent-up bars system not interacting with concrete. Frontiers of Structural and Civil Engineering, 13(6), pp.1301-1315.

*Presenting innovative ensemble model for prediction of the load carrying capacity of composite castellated steel beam under fire. In Structures (Vol. 33, pp. 4031-4052). Elsevier.

*Prediction of the load-carrying capacity of reinforced concrete connections under post-earthquake fire. Journal of Zhejiang University-Science A, 2021, 22(6), pp.441-466.

*Computational predictions for estimating the maximum deflection of reinforced concrete panels subjected to the blast load. International Journal of Impact Engineering, 2020, 139, p.103527.

*Surrogate models for the prediction of damage in reinforced concrete tunnels under internal water pressure. Journal of Zhejiang University-SCIENCE A, 22(8), 2021, pp.632-656.

9- The theoretical part of the paper can be improved. The background and description of theoretical modes should be better explained. Graphs and flowcharts may help in this section.

10- Some suggestions for the future works should be proposed in the conclusion.

11-The conclusion should be rewritten. The authors should add important results and should explain the summary of their study. Please add the main findings.

6. PLOS authors have the option to publish the peer review history of their article (what does this mean?). If published, this will include your full peer review and any attached files.

Reviewer #1: No

Reviewer #2: No

Reviewer #3: No

---

## [Author Response · Author response to Decision Letter 0]

10 May 2022

I have submitted a rebuttal letter with detailed responses to all comments from the academic editors and reviewers.

---

## [Decision Letter · Decision Letter 1]

26 May 2022

PONE-D-22-04046R1Performance of a prestressed efficiently prefabricated beam-column connectionPLOS ONE

Dear Dr. Jiang,

Thank you for submitting your manuscript to PLOS ONE. After careful consideration, we feel that it has merit but does not fully meet PLOS ONE’s publication criteria as it currently stands. Therefore, we invite you to submit a revised version of the manuscript that addresses the points raised during the review process.

Please, address all the comments made by the reviewers. More explanations regarding the novelty, principles of operation, methodology of loading, etc. are required as stated by the reviewers.  Please submit your revised manuscript by Jul 10 2022 11:59PM. If you will need more time than this to complete your revisions, please reply to this message or contact the journal office at plosone@plos.org. Please include the following items when submitting your revised manuscript:A rebuttal letter that responds to each point raised by the academic editor and reviewer(s). You should upload this letter as a separate file labeled 'Response to Reviewers'.A marked-up copy of your manuscript that highlights changes made to the original version. You should upload this as a separate file labeled 'Revised Manuscript with Track Changes'.An unmarked version of your revised paper without tracked changes. You should upload this as a separate file labeled 'Manuscript'.If applicable, we recommend that you deposit your laboratory protocols in protocols.io to enhance the reproducibility of your results. Protocols.io assigns your protocol its own identifier (DOI) so that it can be cited independently in the future. For instructions see: https://journals.plos.org/plosone/s/submission-guidelines#loc-laboratory-protocols. Additionally, PLOS ONE offers an option for publishing peer-reviewed Lab Protocol articles, which describe protocols hosted on protocols.io. Read more information on sharing protocols at https://plos.org/protocols?utm_medium=editorial-email&utm_source=authorletters&utm_campaign=protocols.

We look forward to receiving your revised manuscript.

Kind regards,

Antonio Riveiro Rodríguez, PhD

Academic Editor

PLOS ONE

Journal Requirements:

Reviewers' comments:

Reviewer's Responses to Questions

**Comments to the Author**

1. If the authors have adequately addressed your comments raised in a previous round of review and you feel that this manuscript is now acceptable for publication, you may indicate that here to bypass the “Comments to the Author” section, enter your conflict of interest statement in the “Confidential to Editor” section, and submit your "Accept" recommendation.

Reviewer #1: (No Response)

Reviewer #2: (No Response)

Reviewer #3: All comments have been addressed

2. Is the manuscript technically sound, and do the data support the conclusions?

Reviewer #1: Partly

Reviewer #2: Yes

Reviewer #3: Yes

3. Has the statistical analysis been performed appropriately and rigorously? 

Reviewer #1: N/A

Reviewer #2: Yes

Reviewer #3: Yes

4. Have the authors made all data underlying the findings in their manuscript fully available?

Reviewer #1: Yes

Reviewer #2: Yes

Reviewer #3: Yes

5. Is the manuscript presented in an intelligible fashion and written in standard English?

Reviewer #1: Yes

Reviewer #2: Yes

Reviewer #3: Yes

6. Review Comments to the Author

Reviewer #1: Please provide references to my comments 2 and 3

Comment 2: - It seems that the shear force of the beam is resisted only by the steel bars. Please

provide more information on the design of linking bars at the beam-column interface.

Whether the steel bars are sufficiently strong and stiff to resist the shear force?

Comment 3: - Figure 5: the energy dissipation in the positive loading direction is much larger than

in the negative direction, which is due to the use of energy consuming steel bars at the

top region of the beam. Please justify such treatment in the text.

Reviewer #2: A new prestressed efficiently prefabricated beam-column connection is proposed in this study, and a nice experimental study was carried out. The work is technically sound and is worth publication. The reviewer also appreciates the authors’ effort in improving the language of the paper. Before the paper can be formally accepted, some technical issues need to be further addressed:

1. The authors are suggested to highlight the scientific importance and novelty of the paper, as this connection may be only suited to a certain type of structures in a certain country, and may lack universality. For example, is PPEFF system used in other countries apart from China?

2. The full name of the term PPEFF should be provided again when it first appears in the main text, even though an explanation has been given in the abstract.

3. The basic working principle of a PPEFF connection should be explained before introducing the test program.

4. Any gap opening expected between the beam and the column? The authors may be aware that gap opening mechanism leads to detrimental “beam-growth” or “frame expansion” effect, and as a result severe damage to the slab system. The authors should discuss or at least recognize this issue. For more details about frame-expansion, the authors may refer to (and mention): Experimental and numerical studies on self-centring beam-to-column connections free from frame expansion. Engineering Structures, 198, 109526.

5. The rationale behind the loading protocol (sequence) should be briefly discussed, and the authors should refer to (and mention) the following work: Loading protocols for experimental seismic qualification of members in conventional and emerging steel frames. Earthquake Engineering & Structural Dynamics, 49(2), 155-174, for more details about how a reasonable loading protocol is developed for quasi-static testing of structural members.

6. The hysteretic loop is not symmetrical. This is a detrimental effect, and may not be accepted by the community of civil engineers. Please comment on this issue.

Reviewer #3: The manuscript was revised carefully based on the comments. I strongly recommend the paper for publication.

7. PLOS authors have the option to publish the peer review history of their article (what does this mean?). If published, this will include your full peer review and any attached files.

Reviewer #1: No

Reviewer #2: No

Reviewer #3: **Yes: **Aydin Shishegaran

---

## [Author Response · Author response to Decision Letter 1]

11 Jun 2022

Reviewer #1: Please provide references to my comments 2 and 3

Comment 2: - It seems that the shear force of the beam is resisted only by the steel bars. Please

provide more information on the design of linking bars at the beam-column interface.

Whether the steel bars are sufficiently strong and stiff to resist the shear force?

Comment 3: - Figure 5: the energy dissipation in the positive loading direction is much larger than

in the negative direction, which is due to the use of energy consuming steel bars at the

top region of the beam. Please justify such treatment in the text.

Reviewer #2: A new prestressed efficiently prefabricated beam-column connection is proposed in this study, and a nice experimental study was carried out. The work is technically sound and is worth publication. The reviewer also appreciates the authors’ effort in improving the language of the paper. Before the paper can be formally accepted, some technical issues need to be further addressed:

1. The authors are suggested to highlight the scientific importance and novelty of the paper, as this connection may be only suited to a certain type of structures in a certain country, and may lack universality. For example, is PPEFF system used in other countries apart from China?

2. The full name of the term PPEFF should be provided again when it first appears in the main text, even though an explanation has been given in the abstract.

3. The basic working principle of a PPEFF connection should be explained before introducing the test program.

4. Any gap opening expected between the beam and the column? The authors may be aware that gap opening mechanism leads to detrimental “beam-growth” or “frame expansion” effect, and as a result severe damage to the slab system. The authors should discuss or at least recognize this issue. For more details about frame-expansion, the authors may refer to (and mention): Experimental and numerical studies on self-centring beam-to-column connections free from frame expansion. Engineering Structures, 198, 109526.

5. The rationale behind the loading protocol (sequence) should be briefly discussed, and the authors should refer to (and mention) the following work: Loading protocols for experimental seismic qualification of members in conventional and emerging steel frames. Earthquake Engineering & Structural Dynamics, 49(2), 155-174, for more details about how a reasonable loading protocol is developed for quasi-static testing of structural members.

6. The hysteretic loop is not symmetrical. This is a detrimental effect, and may not be accepted by the community of civil engineers. Please comment on this issue.

Reviewer #3: The manuscript was revised carefully based on the comments. I strongly recommend the paper for publication.

---

## [Decision Letter · Decision Letter 2]

22 Jun 2022

Performance of a prestressed efficiently prefabricated beam-column connection

PONE-D-22-04046R2

Dear Dr. Jiang,

We’re pleased to inform you that your manuscript has been judged scientifically suitable for publication and will be formally accepted for publication once it meets all outstanding technical requirements.

Kind regards,

Antonio Riveiro Rodríguez, PhD

Academic Editor

PLOS ONE

Reviewers' comments:

Reviewer's Responses to Questions

**Comments to the Author**

1. If the authors have adequately addressed your comments raised in a previous round of review and you feel that this manuscript is now acceptable for publication, you may indicate that here to bypass the “Comments to the Author” section, enter your conflict of interest statement in the “Confidential to Editor” section, and submit your "Accept" recommendation.

Reviewer #1: All comments have been addressed

Reviewer #2: All comments have been addressed

Reviewer #3: All comments have been addressed

2. Is the manuscript technically sound, and do the data support the conclusions?

Reviewer #1: Yes

Reviewer #2: Yes

Reviewer #3: Yes

3. Has the statistical analysis been performed appropriately and rigorously? 

Reviewer #1: N/A

Reviewer #2: Yes

Reviewer #3: Yes

4. Have the authors made all data underlying the findings in their manuscript fully available?

Reviewer #1: Yes

Reviewer #2: Yes

Reviewer #3: Yes

5. Is the manuscript presented in an intelligible fashion and written in standard English?

Reviewer #1: Yes

Reviewer #2: Yes

Reviewer #3: Yes

6. Review Comments to the Author

Reviewer #1: The authors have made efforts in addressing my comments. I am happy with the revisions. The paper can now be accepted.

Reviewer #2: (No Response)

Reviewer #3: The manuscript was revised based on the comments. I recommend this version of the manuscript for publication.

7. PLOS authors have the option to publish the peer review history of their article (what does this mean?). If published, this will include your full peer review and any attached files.

Reviewer #1: No

Reviewer #2: No

Reviewer #3: **Yes: **Aydin Shishegaran

---

## [Editor Report · Acceptance letter]

6 Jul 2022

PONE-D-22-04046R2 

Performance of a prestressed efficiently prefabricated beam-column connection 

Dear Dr. Jiang:

I'm pleased to inform you that your manuscript has been deemed suitable for publication in PLOS ONE. Congratulations! Your manuscript is now with our production department. 

Kind regards, 

on behalf of

Dr. Antonio Riveiro Rodríguez 

Academic Editor

PLOS ONE